# Single-cell multi-omics analysis of human testicular germ cell tumor reveals its molecular features and microenvironment

Xiaojian Lu[1,2,3,10], Yanwei Luo[4,10], Xichen Nie[5,10], Bailing Zhang[1,2,3], Xiaoyan Wang [1,2], Ran Li[1,2], Guangmin Liu[6], Qianyin Zhou[6], Zhizhong Liu[6], Liqing Fan[6,7], James M. Hotaling[5], Zhe Zhang [8,9] ✉, Hao Bo [6,7] ✉ & Jingtao Guo [1,2,3] ✉

Seminoma is the most common malignant solid tumor in 14 to 44 year-old men. However, its molecular features and tumor microenvironment (TME) is largely unexplored. Here, we perform a series of studies via genomics profiling (single cell multi-omics and spatial transcriptomics) and functional examination using seminoma samples and a seminoma cell line. We identify key gene expression programs share between seminoma and primordial germ cells, and further characterize the functions of TFAP2C in promoting tumor invasion and migration. We also identify 15 immune cell subtypes in TME, and find that subtypes with exhaustion features were located closer to the tumor region through combined spatial transcriptome analysis. Furthermore, we identify key pathways and genes that may facilitate seminoma disseminating beyond the seminiferous tubules. These findings advance our knowledge of seminoma tumorigenesis and produce a multi-omics atlas of in situ human seminoma microenvironment, which could help discover potential therapy targets for seminoma.

Testicular germ cell tumors (TGCTs) are the most common malignant solid tumor among men ages 14–44, constituting up to 95% of all testicular cancers[1–3]. They are characterized by cell of origin[4]. Seminoma is the most common type of TGCTs, accounting for about 60%[5,6], and is considered to originate from germ cell neoplasia in situ (GCNIS)[7–9].

The human germline derives from progenitors called primordial germ cells (PGCs), which are specified during pre-implantation of human development[10,11]. PGCs migrate into the genital ridge 4–5 weeks after fertilization and differentiate into advanced germ cells through intricate interactions with the niche[12,13]. The origin of TGCTs is thought to be developmentally blocked PGCs or gonocytes through epidemiological and histological studies[14–17].

Key transcription factors (TFs) such as TFAP2C, SOX17, OCT4/POU5F1 and NANOG are highly expressed in PGCs, and consequently are excellent diagnostic markers for seminoma and GCNIS[14,18,19]. TFAP2C and SOX17 are key factors which activate the germline specification program, whereas POU5F1 and NANOG are the main

[1]State Key Laboratory of Stem Cell and Reproductive Biology, Institute of Zoology, Chinese Academy of Sciences, Beijing, China. [2]Beijing Institute for Stem Cell and Regenerative Medicine, Beijing, China. [3]University of Chinese Academy of Sciences, Beijing, China. [4]Department of Blood Transfusion, The Third Xiangya Hospital of Central South University, Changsha, China. [5]Division of Urology, Department of Surgery, University of Utah School of Medicine, Salt Lake City, UT, USA. [6]NHC Key Laboratory of Human Stem Cell and Reproductive Engineering, Institute of Reproductive and Stem Cell Engineering, School of Basic Medical Science, Central South University, Changsha, Hunan, China. [7]Clinical Research Center for Reproduction and Genetics in Hunan Province, Reproductive and Genetic Hospital of CITIC-Xiangya, Changsha, Hunan, China. [8]Department of Urology, Peking University Third Hospital, Beijing, China. [9]Center for Reproductive Medicine, Peking University Third Hospital, Beijing, China. [10]These authors contributed equally: Xiaojian Lu, Yanwei Luo, Xichen Nie. ✉e-mail: zhezhang@bjmu.edu.cn; bohao1990@163.com; jingtao.guo@ioz.ac.cn

pluripotency factors[13,20–23]. As germ cells differentiate into gonocytes or pro-spermatogonia, these factors become downregulated. Interestingly, in seminoma tumors from adult men, these genes also display strong expression. However, the roles of those factors in seminoma and their impact on tumor progression still remains unknown.

Interactions between tumor cells and immune cells are a core focus of cancer research. A comprehensive analysis of immune cell infiltration in tumors can provide a deeper understanding of the mechanisms underlying tumor survival, and may lead to potential targets for cancer treatment. Cancer is a systematic disease, accompanied by long-term infiltration of immune cells during tumor progression. Initially, immune cells recruited to the tumor region are thought to play an agonist role. However, as tumor progresses, the immune landscape in the tumor microenvironment (TME) alters, and immune cells differentiate into different states, either hindering or supporting the growth of tumor cells[24–26]. Therefore, a detailed analysis of the immune cell status in the seminoma TME can provide better understanding of this tumor progression and identify feasible targets for immunotherapy. Previous studies on primary seminoma have shown that there are high levels of T cell receptor (TCR) and B cell receptor (BCR) signals within seminoma, indicating the presence of infiltrated immune cells[27]. A recent study involving single-cell sequencing (scRNA-seq) of metastatic seminoma identified the presence of immune cells in different states, further demonstrating the complex diversity of immune cells in the seminoma microenvironment[6]. In vitro co-culture experiments of a seminoma cell line (TCam-2) with immune cells also revealed frequent interactions between tumor cells and immune cells[28]. All of these results demonstrate the importance of immune cells in the development of seminoma. However, how the immune environment of in situ seminoma interact with tumor cells remains unknown. Therefore, it is necessary to investigate the immune microenvironment of primary seminomatous tumors.

As one of the most curable solid malignancies, with excellent >95% 5 year survival[29], the investigation of seminoma is relatively limited. However, about 15-30% of seminoma patients may relapse after first-line chemotherapy[30,31]. Moreover, since the germ cells are also highly sensitive to chemotherapy, chemotherapy has substantial impacts on fertility; with many seminoma patients becoming azoospermic after chemotherapy and many remaining permanently hypogonadal[32]. Considering that most patients with seminoma are at their reproductive age, fertility preservation is essential for them. Although patients are recommended to undertake semen cryopreservation before treatment, many do not due to their age or advanced disease state[33].

A more targeted therapeutic treatment is needed to reduce relapse and cause less gonadotoxicity. Cutting edge high-throughput genomics sequencing techniques now provide new avenues to understand the normal developmental process as well as testis cancer pathogenesis. We previously explored the molecular features of human testis development, as well as male infertility[12,34–36].

In this work, to better understand the molecular features of seminoma and tumor-niche interactions, we perform a series of experiments using genomics profiling (single cell multi-omics and spatial transcriptomics) on human seminoma samples and Cleavage Under Target & Tagmentation (CUT&Tag) on a human seminoma cell line. Our study indicates that seminoma shares a more similar gene expression pattern with PGCs, and TFAP2C can promote the invasion and metastasis of tumor cells. Moreover, our work reveals spatial heterogeneity in the distribution of T cells with different functional states in tumors, while macrophages in seminoma can participate in extracellular matrix degradation through the secretion of MMP9 and CTSK. Our work provides further information for understanding seminoma tumorigenesis and supply a comprehensive multi-omics dataset for further research, and also provides data support

for the development of additional potential therapeutic strategies of seminoma.

## Results

### Single-cell transcriptome profiling of seminoma

We isolated single cells of tumor tissues from 4 adult men and performed scRNA-seq using 10x Genomics platform (Fig. 1a). From a total of ~18,746 cells, 10,817 passed standard quality control (QC) dataset filters and were retained for downstream analysis (see "Method"). We obtained ~37,629-171,789 reads/cell, which enabled the analysis of ~1370–6796 genes/cell.

We first performed UMAP (uniform manifold approximation and projection) for dimension reduction analysis on the combined datasets using the Seurat package (Fig. 1b and Supplementary Fig. 1a)[37]. We found that cells formed clusters independent of donor origin (Supplementary Fig. 1a), indicating minimal batch effect or individual variation. To annotate cell identities, we projected expression of sets of known cell type-specific marker genes and identified 8 major cell types, including seminoma ($POU5F1^+$, $TFAP2C^+$ and $NANOG^+$), T cells ($CD3D^+$ and $CD3E^+$), B cells ($CD79A^+$ and $MZB1^+$), NK cells ($CD160^+$ and $FCGR3A^+$), endothelial cells ($CDH5^+$), macrophage ($CD68^+$ and $LYZ^+$), plasmacytoid dendritic cells (pDCs; $LILRA4^+$), smooth muscle cells ($ACTA2^+$ and $NOTCH3^+$) (Fig. 1b, e; see Supplementary Fig. 1e for additional markers). Through principal component analysis (PCA), we discovered that the same cell types from different samples cluster together, further demonstrating the similarity of our data and the accuracy of cell type definition (Supplementary Fig. 1c). We clustered the marker genes of each cell type and annotated these genes with Gene Ontology (GO) terms (Fig. 1f). We further examined the distribution of cell types in each donor sample. Notably, the largest proportion is immune cells, indicating pronounced immune infiltration (Fig. 1c and Supplementary Fig. 1b). We also performed immunostaining and proved that abundant immune cells in and around seminoma constitute the tumor microenvironment (TME) (Fig. 1d and Supplementary Fig. 1d). Using inferCNV analysis, we found that tumor cells enriched more copy number variations (CNVs) than non-tumor cells (Supplementary Fig. 1f).

### Seminoma and PGCs shared expression of germline specification and pluripotency factors

We next explored the molecular features and possible origin of seminoma by comparing seminoma cells and other male germ cell types. We extracted 375 tumor cells which we identified in Fig. 1b, and then we integrated this dataset with adult health germ cell datasets we previously published (Fig. 2a)[34]. By comparing the gene expression between seminoma and all adult germ cell stages, we found that seminoma cells display highly similar gene expression pattern with spermatogonial stem cells (SSCs) and differentiating spermatogonia (Fig. 2b). Since these two stages are early germ cells in the adult testis, we reasoned that seminoma may originate from earlier germline such as PGCs or gonocytes in the fetal or neonatal testis, which has been proposed by several prior studies[38]. Therefore, we combined seminoma data with data from our previously published work, including those from PGCs, fetal state 0-like cells (state f0)[12], and adult spermatogonial states (state 0–4)[34], all of which are early germ cells (Supplementary Fig. 2a). Through PCA, we found that seminoma and PGCs display the highest similarity (Fig. 2c). This finding was also supported by correlation analysis (Fig. 2d). We further combined and re-clustered tumor dataset with all germ cells datasets mentioned above (Supplementary Fig. 2b), and also conducted correlation analysis among these datasets. As expected, seminoma is most similar to PGCs (Supplementary Fig. 2c).

We next performed clustering analysis and heatmap visualization to examine gene sets specifically expressed in each cluster (Fig. 2e and Supplementary Fig. 2a). As expected, seminoma and PGCs shared

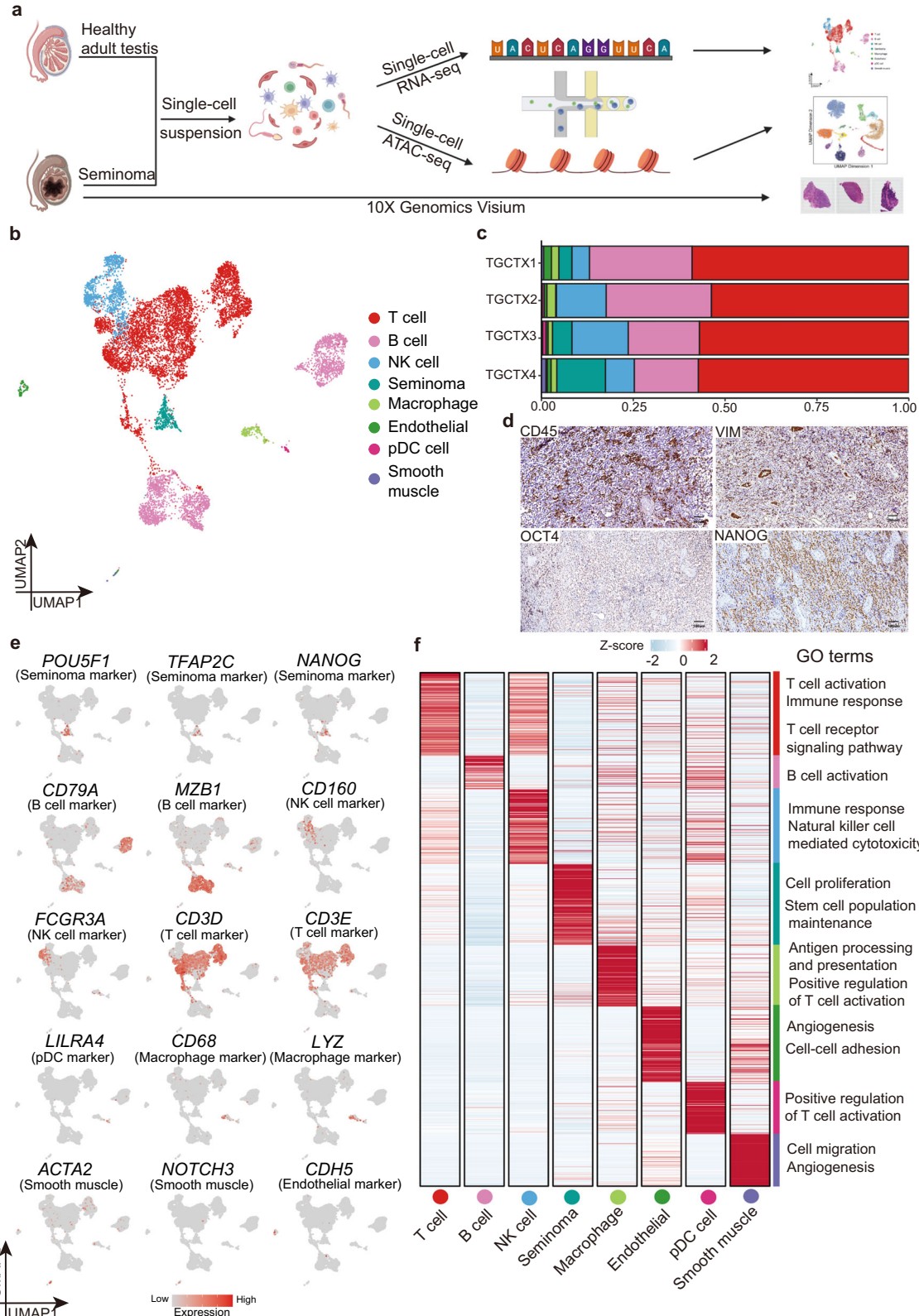

**Fig. 1 | Single-cell transcriptome profiling and analysis of seminoma.**
**a** Schematic illustration of the experimental workflow. This plot created with BioRender.com. **b** Dimension-reduction presentation of the combined single-cell transcriptome data from seminoma samples (*n* = 4), Uniform manifold approximation and projection (UMAP) plot showing the major cell types (*n* = 10,817 cells). Dots represent individual cells, and colors represent different cell populations. **c** Cell ratio of cell types from (**b**). Colors represent different cell populations which

is consistent with Fig. 1b. Source data are provided as a Source data file. **d** IHC staining for seminoma and immune cells in seminoma samples (*n* = 3). Brown color represents positive signal cells. CD45: immune cells; VIM: somatic cells; NANOG and OCT4/POU5F1: seminoma tumor cells. Scale bar, 100 μm. **e** Expression patterns of selected markers projected on the UMAP plot. **f** Heatmap of marker genes in each cluster and GO terms enrichment.

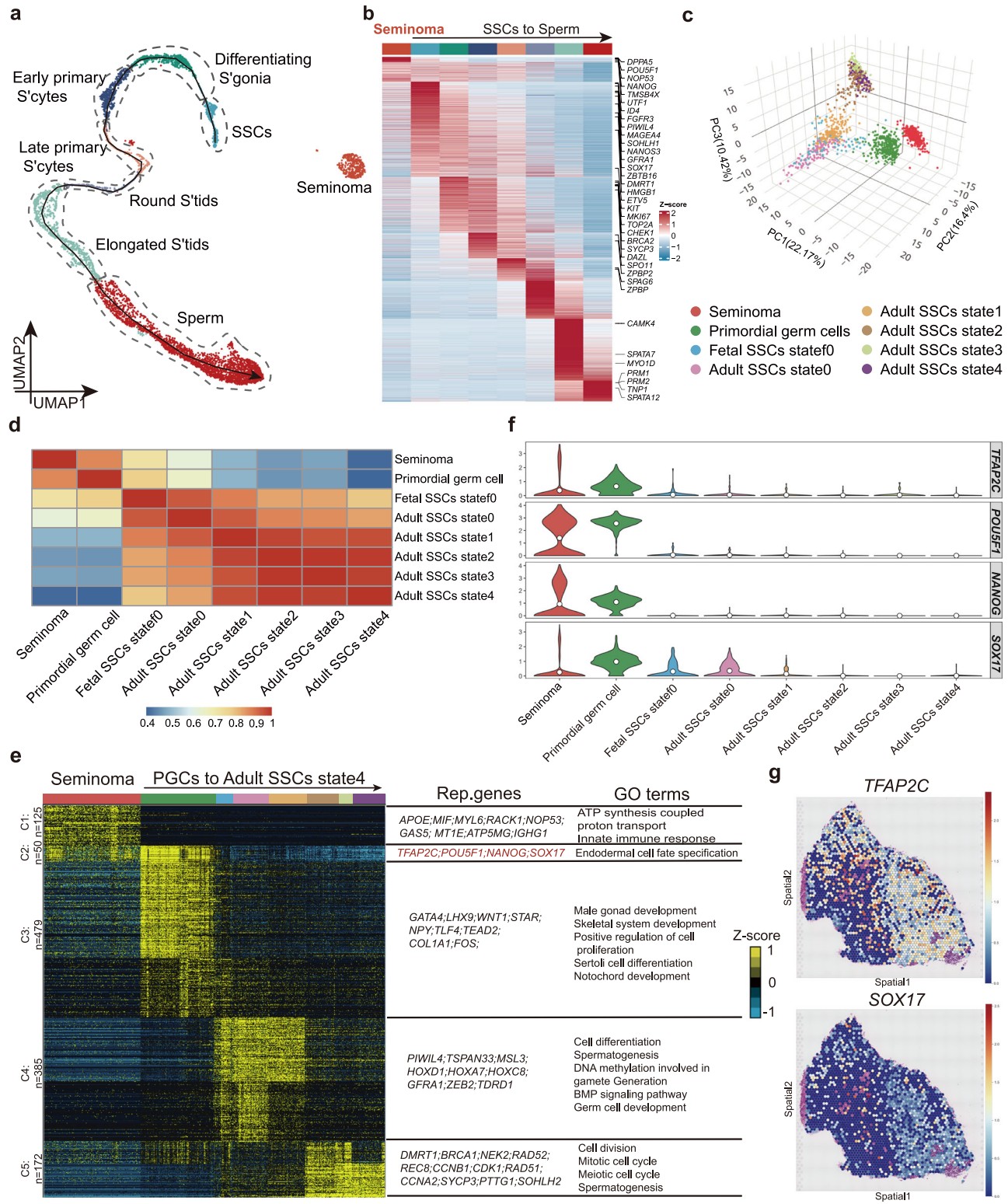

**Fig. 2 | Seminoma and PGCs shared gene expression programs in pluripotency and germ cell development. a** UMAP plot of normal germ cells (*n* = 3 samples, data from GEO:GSE120508[34]) combined with tumor cells (*n* = 5163 cells, 4 samples), curve with arrow represents development trajectory of normal germ cells from SSCs to sperm. **b** Heatmap showing average gene expression patterns of each cell type in (Fig. 2a). **c** 3D PCA of seminoma and early germ cells (UMAP shown in Supplementary Fig. 2a) showing that seminoma and PGCs display highest similarity. **d** Correlation analysis of cell types in (Fig. 2c) calculated using Pearson's correlation coefficients. Source data are provided as a Source data file. **e** K-means clustering of genes exhibiting differential and shared expression between seminoma and all early germ cell stages. Curve with arrow represents developmental timeline of male germ cells. Each row represents a gene, and each column represents a single cell. Shared and differential gene expression levels use a Z score as defined by the color key; associated GO terms (using DAVID version 6.8) are given on the right of the corresponding gene clusters. **f** Expression of *TFAP2C, POU5F1, NANOG* and *SOX17* in cell types in (Fig. 2c). The white dots represent mean expression. **g** *TFAP2C* and *SOX17* expression patterns in spatial transcriptome data.

specific expression of genes (C2) such as *TFAP2C, POU5F1, NANOG* and *SOX17* (Fig. 2e, f). All of them encoded key TFs involved in pluripotency regulation during normal development and expressed in embryonic stem cells and PGCs[10,13,20,39].

To validate the expression patterns of these key genes in seminoma, we performed spatial transcriptomics profiling (using the 10x Visium platform) with tumor tissue sections from three seminoma patients, and obtained about 6403 spots and 53,829 genes for downstream analysis (Supplementary Fig. 2e, f). We found *TFAP2C* and *SOX17* also highly expressed within the areas of seminoma in the spatial data (Fig. 2g and Supplementary Fig. 2f). Moreover, seminoma displayed high expression of genes associated with innate immune response, regulation of B cell activation (e.g., *CCL5, APOE, MIF, NOP53*), suggesting that the immune activity in TME play a role in tumorigenesis. We also observed that genes involved in gonad development and stem cell fate specification specifically expressed in PGCs (Supplementary Fig. 2d), consistent with our prior work[12]. Taken together, our results suggested that human seminoma and PGCs share expression patterns in key pluripotency and developmental genes including *TFAP2C, POU5F1, SOX17* and *NANOG*. This promoted studies to further investigation of the exact functions of these genes in seminoma.

### Binding motifs of key TFs enriched in seminoma revealed by scATAC-seq analysis

To gain more mechanistic insights into the roles of key TFs in seminoma development, we profiled the chromatin accessibility landscapes of a seminoma sample from a tumor patient (3 replicates) and a testicular sample from a healthy control donor (2 replicates) by single-cell assay for transposase accessible chromatin sequencing (scATAC-seq). After filtering out low-quality cells and doublets, we obtained 24, 005 cells with high-quality chromatin open sites for downstream analysis.

We combined scATAC-seq data from the tumor sample with these from the control sample and performed dimension reduction and clustering analysis using ArchR (Fig. 3b)[40]. Here, we identified 11 clusters which belongs to 5 cell types in the combined dataset (Fig. 3a). Given promoter openness is highly correlated with the gene expression level, we annotated cell types using the chromatin openness at the promoter regions of known marker genes (referred to as "gene scores"), which has been used as a prediction of gene expression level based on the accessibility of regulatory elements in the vicinity of the gene[40]. We found that gene scores of key marker genes are highly specific to defined clusters and sufficient to annotate all major cell types from both seminoma and testicular tissues (Fig. 3c and Supplementary Fig. 3a).

To further compare seminoma and germ cells from healthy control, we partitioned out tumor cells and germ cells for a more refined analysis. For a better cell type annotation, we integrated scRNA-seq data with the scATAC-seq data. Here, we obtained 8 refined cell clusters, including a seminoma cluster and 7 normal germ cell clusters, consisting of a developmental trajectory from SSCs through spermatocytes to sperm (Supplementary Fig. 3b and Supplementary Fig. 3c). Next, we sought to identify cell type specific peaks using MACS2 (Supplementary Fig. 3d), which allowed us to examine the key TFs enriched in each cluster based on their motifs (Fig. 3d). Notably, TFAP2C and POU5F1 were highly enriched in seminoma (Fig. 3d and Supplementary Fig. 3e, f), re-enforcing our findings based on scRNA-seq that these key TFs may play an important role in seminoma. A set of NFY family factors and HOX factors were found as SSC enriched TFs (Fig. 3d), which coincided with our previously work[41]. To further validate the protein expression levels of TFAP2C, we performed TFAP2C immunohistochemistry (IHC) staining using seminoma samples from four patients (Fig. 3e), and found high expression of TFAP2C in all the samples, which further proved the reliability of our findings.

### TFAP2C reinforces the migration and invasion programs of seminoma

Given TFAP2C was identified as the top TFs specific to seminoma through both gene expression and chromatin accessibility analysis, we next sought to explore the functional impact of TFAP2C in seminoma. We silenced *TFAP2C* expression in seminoma cell line TCam-2 (Fig. 3f). Compared to control (siNC), *TFAP2C* expression was reduced by more than 50% in the silenced group (si*TFAP2C*) (Fig. 3g and Supplementary Fig. 3g). We then examined the ability of migration and invasion of tumor cells upon *TFAP2C* knockdown. We observed tumor migration and invasion were both significantly decreased in the *TFAP2C* knockdown group (Fig. 3h). We also found that the proliferation of seminoma was reduced in *TFAP2C* knockdown group (Supplementary Fig. 3h, i). These results suggested that *TFAP2C* promote seminoma migration and invasion, and positively regulate the growth of tumor cells.

To further explore the molecular mechanism underlying how *TFAP2C* promotes seminoma development, we performed RNA-seq profiling on the silenced group and control. Through differential gene expression analysis, we identified upregulated genes ($n = 498$) and downregulated genes ($n = 843$) upon *TFAP2C* silencing (Fig. 3i), indicating *TFAP2C* mainly act as a gene activator in seminoma. Consistently, GO analysis revealed enrichment for terms such as cell adhesion and cell migration in downregulated genes.

We next performed CUT&Tag to capture genes bound and regulated by TFAP2C in the TCam-2 cells. We obtained TFAP2C directly activated genes ($n = 346$) by intersecting the downregulated genes in *TFAP2C* silenced group ($n = 843$) and the TFAP2C binding genes from CUT&Tag ($n = 4906$) (Fig. 3j). Not surprisingly, GO terms such as cell adhesion and migration were enriched (Fig. 3j). Genes associated with cell migration (e.g., *FOSL1, LYPD3, ITGA6*) were identified as TFAP2C targets which are bound by TFAP2C and display reduced expression upon *TFAP2C* knockdown (Fig. 3k). Genes directly activated by TFAP2C are also enriched in extracellular matrix, which is a major component of the cellular microenvironment and influences cell behaviors such as migration[42,43]. Notably, *POU5F1* was directly activated by TFAP2C (Supplementary Fig. 3g). Altogether, these results revealed the key role of TFAP2C in promoting migration and invasion by directly regulating genes such as *FOSL1, LYPD3* and *ITGA6* in the seminoma cell line, promoting further study to understand its role in seminoma tumorigenesis.

### Immune cell infiltration in seminoma

As major cell types identified in the tumor samples are immune cells, we next analyzed immune cell clusters from Fig. 1b to explore the functions of immune cells in the seminoma microenvironment (Fig. 4a). We re-clustered T cells and obtained 5 CD8+ and 4 CD4+ clusters, and 3 B cell clusters, each with its unique signature genes (Supplementary Fig. 4a). We observed high expression levels of naïve marker genes such as *CCR7* and *SELL* in the CD8+ and CD4 + CCR7+ Tn cluster. By analyzing gene expression, we identified effector memory cells (Tem) that exhibited high levels of cytotoxicity-related genes such as *GZMK, GZMA, FCGR3A* and *FGFBP2*. Additionally, we identified the CX3CR1+ Temra cluster (terminally differentiated effector memory or effector cells) with highly expressed genes *CX3CR1, TBX21* and *GZMH*. PDCD1+ Tex (exhaustion) cluster highly expressed exhaustion markers *CTLA4, PDCD1* and *HAVCR2*, representing exhausted CD8 + T cells. These exhaustion markers were also expressed in CD4 + FOXP3+ Treg (regulatory T cells) cluster and CD4 + CXCL13+ Tex cluster. The latter cluster also expressed T follicular helper (Tfh)-like cell markers such as *IL21, BCL6* and *ICOS*. Our findings are consistent with prior research[25,44–46]. Moreover, we identified a CD8+ cluster that expressed *MKI67* and *TOP2A*, indicating that the T cells in this cluster were proliferative (Fig. 4b and Supplementary Fig. 4a)[47].

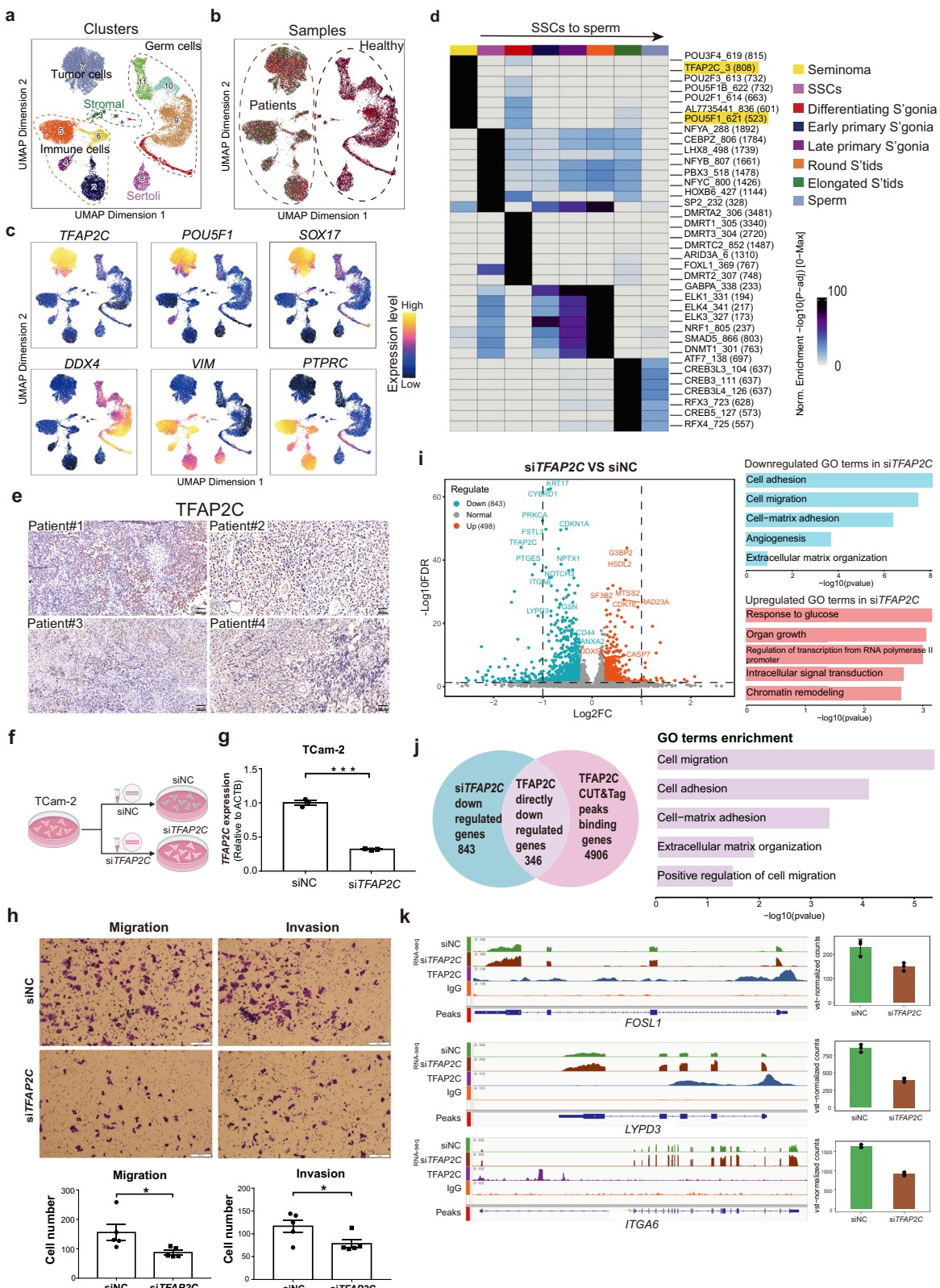

We next integrated scATAC-seq and scRNA-seq data of immune cells to further understand the roles of key TFs in TME (Supplementary Fig. 4b). We found binding motifs for RUNX2, EOMES and ETS1 were enriched in NK cells[48,49]. Binding motifs for RUNX and EOMES were enriched in CD8+ and CD4 + T cells, consistent with prior studies[46,50–52]. To further understand T cell subtypes in the microenvironment, we calculated the proportions of these subtypes (Fig. 4c). Interestingly, we

noted a higher proportion of cells with exhaustion features than those with cytotoxicity, suggesting that the seminoma microenvironment is in an immunosuppressed state. We then applied Monocle algorithm to order CD8 + T cells and CD4 + T cells in pseudotime to infer their developmental trajectories (Fig. 4d and Supplementary Fig. 4c). The pseudotime of CD8 + T cells commenced with CCR7+ Tn and progressed to GZMK+ Tem before culminating with PDCD1+ Tex cells. The

**Fig. 3 | TFAP2C regulates genes associated with migration and invasion of seminoma. a** Dimension-reduction presentation of clusters and cell types annotation in scATAC-seq data from seminoma samples and healthy testicular control samples (*n* = 24,005 cells). **b** Samples origin of data in (Fig. 3b). **c** Predicted gene expression of selected markers in scATAC-seq data using ArchR. *TFAP2C, POU5F1, SOX17*: seminoma tumor cell markers; *DDX4*: germ cells marker; *VIM*: somatic cells marker; *PTPRC*: immune cells marker. **d** TF binding motifs enriched in seminoma tumor cells and adult normal germ cells. Top 7 TFs in each cell types were shown here. **e** IHC staining of TFAP2C in seminoma samples from four patients (*n* = 4). Scale bar, 50 μm. **f** Schematic diagram of knocking down *TFAP2C* in seminoma cell line TCam-2 using small interfering RNA (siRNA). NC: negative control. This plot created with BioRender.com. **g** Barplot showing the expression of *TFAP2C* in si*TFAP2C* and siNC group in TCam-2 cells (*n* = 3 experiments replicates). siNC: control group; si*TFAP2C*: *TFAP2C* were knocked down. *P* value was calculated by two-sided Wilcoxon rank-sum test. ***P* < 0.001. Data are presented as "mean values ±SEM" as appropriate. **h** Crystal violet staining showed migration and invasion of TCam-2 cells in si*TFAP2C* group and siNC group (top). Statistics of migration and invasion cell numbers compared between two groups (below) (*n* = 5). *P* value was calculated by two-sided Wilcoxon rank-sum test. **P* < 0.05. Data are presented as "mean values ± SEM" as appropriate. Source data are provided as a Source data file. Scale bar, 200 μm. **i** Volcano plot displays differential gene expression in si*TFAP2C* compared with siNC. The number represents gene count of downregulated (*n* = 843) and upregulated (*n* = 498). GO terms enrichment in downregulated genes and upregulated genes are shown on the right. The length of the bar represents the value of -log10pvalue. Source data are provided as a Source data file. **j** TFAP2C directly downregulated genes (*n* = 346) through intersecting downregulated genes in si*TFAP2C* group (*n* = 843) with TFAP2C bound genes (*n* = 4906). Go terms enriched in TFAP2C directly downregulated genes are displayed on the right. The length of the bar represents the value of -log10pvalue. **k** Coverage plots of representative genes which are directly downregulated by TFAP2C. The green color represents the siNC group and the brown color represents the siTFAP2C group (*n* = 3). Data are presented as "mean values ±SEM" as appropriate. Source data are provided as a Source data file.

trajectory plot exhibited a distinct separation between the CX3CR1+ Temra and PDCD1+ Tex clusters, indicating functional divergence between these cells. Furthermore, a significant proportion of GZMK+ Tem cells leaned toward exhaustion, while the rest differentiated into CX3CR1+ Temra cells, signifying the dynamic development of the T cell population. Exhausted T cells were predominantly enriched the later stages, demonstrating a transition of T cells from activation to exhaustion in tumor. A similar pattern was also found in the CD4 + T cells (Fig. 4d and Supplementary Fig. 4c).

We also defined three subtypes of B cells in our dataset, including naïve B cells, memory B cells and plasma cells, and observed a clear developmental trajectory from naïve through memory B cells to plasma cells through pseudotime analysis (Supplementary Fig. 4d). Binding motif for PAX5, a deciding factor for B cell specification, was enriched in naïve and memory B cells[53,54]. Motifs for POU2F2 and IRF4 were highly enriched in plasma cells (Supplementary Fig. 4b)[50,55]. GO analysis revealed that pathways associated with antigen presenting were highly enriched in naïve and memory B cells, and "B cell activation" and "humoral immune response" were enriched in plasma cells, consistent with their known functions (Supplementary Fig. 4e)[56,57].

To verify the spatial relationship between T cell subtypes and tumors in seminoma, we integrated scRNA-seq data of T cell subtypes and seminoma with spatial transcriptomics data (Fig. 4e and Supplementary Fig. 4f, g)[58]. Surprisingly, we noticed that both CD8+ and CD4+ cells expressing exhaustion features were located in closer proximity to the tumor cells spatially. In contrast, naïve and effector cells were found further away from the tumor cells. We also found that several key cytotoxic genes (such as *NKG7*, *GNLY*, *PRF1*) were more highly expressed in the non-tumor regions (Supplementary Fig. 4h). These genes were canonical cytolytic effector molecules secreted by effector T cells and NK cells, which may play a role in killing target cells[47,59–61]. As their expression level largely represents the ability of T cells to attack tumor cells, we speculated that T cells located intra-tumoral may have a diminished firepower. IHC staining of CD3 and PDCD1 confirmed our findings that exhaustion genes were more frequently expressed within tumor regions (Fig. 4f). Survival analysis of *FOXP3* and *IL21*, which mainly expressed in dysfunctional T cell subtypes, using TGCA TGCT cohort suggested high expression of these two genes were associated with poor prognosis (Fig. 4g). Altogether, these results indicated that tumor-infiltrating cells in the seminoma microenvironment are in an immunosuppressed state.

## Macrophages may promote seminoma breaking through seminiferous tubule by expressing *MMP9* and *CTSK*

Seminoma is considered to originate from GCNIS, which are located inside the seminiferous tubules[62]. Therefore, seminoma progression requires tumor cells to break through the seminiferous tubules to proliferate and expand, while the detailed molecular mechanism is largely unexplored (Fig. 5a). We utilized spatial data to identify pathways enriched in cells located in the tumor margin region, and found "Degradation of the extracellular matrix" were highly expressed in all samples (Fig. 5b). By examining key genes within this pathway, we found, *MMP9*, which encodes a zinc-dependent endopeptidase that can promote tumor growth by degrading matrix barriers, highly expressed in macrophages (Fig. 5c). Cathepsin K (encoded by *CTSK*), a lysosomal cysteine proteinase involved in bone remodeling and tumor invasiveness[63,64], is known to cleave and activate *MMP9* in tumors. Notably, both *MMP9* and *CTSK* expression displayed positive correlation with macrophages region in spatial data (Fig. 5d and Supplementary Fig. 5a).

To further examine the functions of *MMP9* and *CTSK* in seminoma, we subset immune cells (*PTPRC*+), and explored the expression patterns of markers for T cells (*CD3D*+ and *CD3E*+) and B cells (*CD79B*+; *MS4A1*+ and *SDC1*+) (Supplementary Fig. 5c). Consistently, we found that *MMP9* and *CTSK* displayed high expression in macrophages (*CD68*+ *and LYZ*+) (Fig. 5e). Survival analysis revealed that high expression of *MMP9* and *CTSK* lead to poor prognosis (Fig. 5f). To further validate, we performed Immunofluorescence (IF) staining for MMP9 and CTSK in the tumor patient tissues and found that both MMP9 and CTSK had a co-localization with CD68 (Fig. 5g). Altogether, our analysis provided lines of evidence to support the hypothesis that macrophages may express MMP9 and CTSK in order to help tumor cell growth and break through the seminiferous tubules.

## Seminoma interacts with immune cells through MIF signal pathway in TME

To explore the intercellular communication in seminoma, we performed CellChat analysis (Supplementary Fig. 6a)[65]. We found that tumor cells primarily affected immune cell types via MIF-(CD74 + CXCR4) ligand-receptor pairs (Fig. 6a). We found that in MIF (macrophage migration inhibitory factor) pathway, tumor cells mainly act as senders (Fig. 6b and Supplementary Fig. 6b, c). Upon examining expression of the MIF pathway genes in all cell types, we found that *MIF* had the highest expression in seminoma, while its receptors genes were rarely expressed in tumors (Fig. 6c). This indicated that tumor cells interact with other cells mainly through the MIF signaling pathway but rarely affect themselves. Previous study has suggested that *MIF* display high expression in various tumor types such as breast cancer[66], and it has been reported that immune cells in response to tumor-derived MIF may play a tumor-promoting role[67,68]. Therefore, we hypothesized that tumor cells may secrete MIF to repress immune cell functions in the seminoma microenvironment.

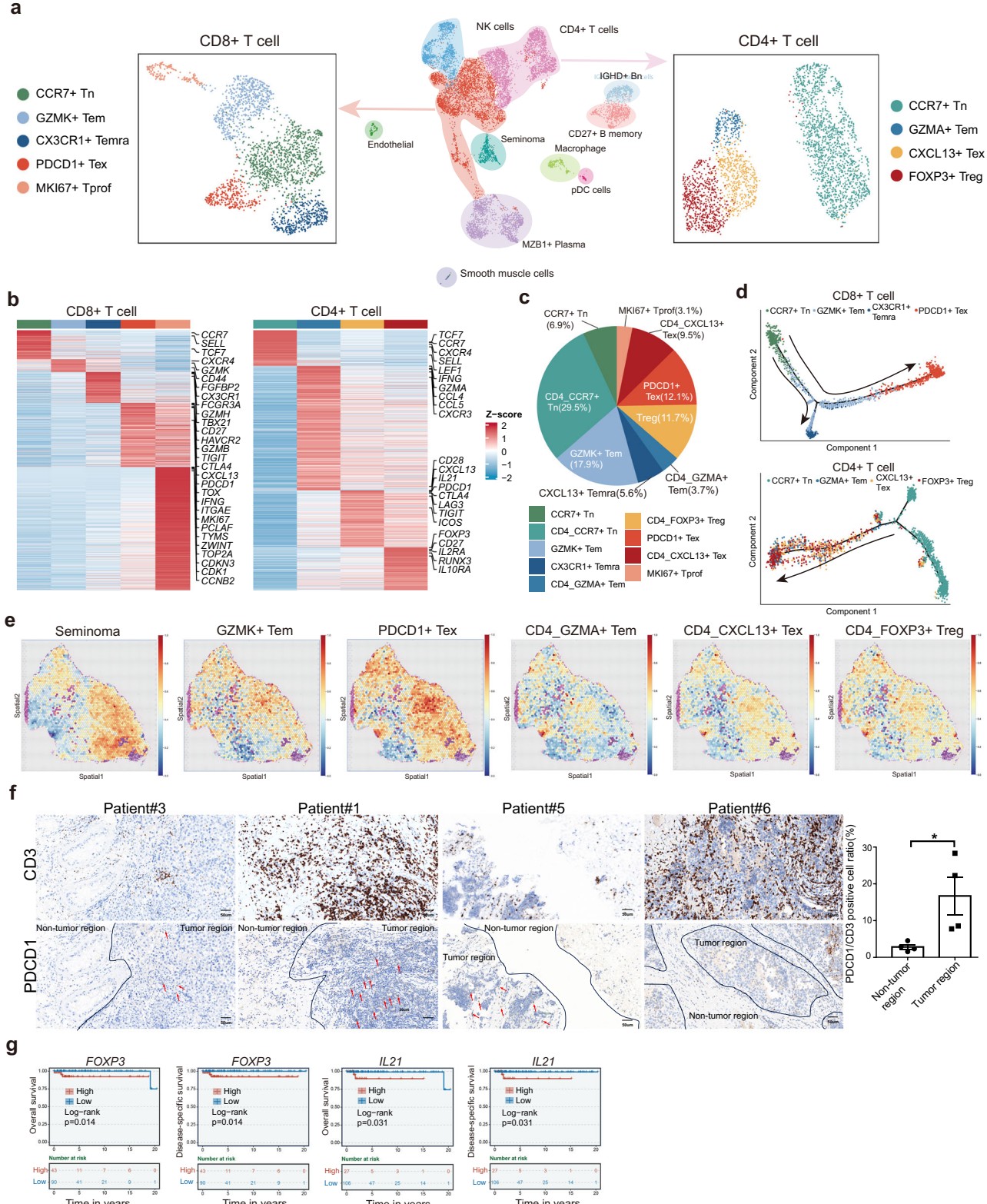

**Fig. 4 | T cell subtypes analysis. a** UMAP plot of immune cells subtypes (*n* = 10,672 cells). Re-clustering of CD8 + T cells (left, *n* = 2382 cells) and CD4 + T cells (right, *n* = 2840 cells). The number of samples is 4. **b** Heatmap of marker genes expressed in CD8+ and CD4 + T cells. The color above the heatmap corresponds to the clusters in UMAP show in (**a**). **c** The cell ratio of all CD8+ and CD4 + T cells. Source data are provided as a Source data file. **d** Trajectories of CD8+ (upper) and CD4+ (below) T cells from pseudotime analysis. **e** Spatial distribution of tumor cells, partial subgroups of CD8 + T cells and CD4 + T cells. **f** IHC staining validated the expression of CD3 and PDCD1 in the tumor regions and non-tumor regions in four

seminoma samples. The bar plot represented the cell ratio of PDCD1+ cells (dysfunctional cells). The white color represents the PDCD1+/CD3+ cell ratio in non-tumor region and the black color represents the PDCD1+/CD3+ cell ratio in tumor region. *P* value was calculated by two-sided Wilcoxon rank-sum test. *\**P* < 0.05. Data are presented as "mean values ±SEM" as appropriate. Source data are provided as a Source data file. Scale bar, 50 μm. **g** Survival analysis showing that *FOXP3 and IL21* high expression was associated with worse overall survival (OS) and disease-specific survival in the TCGA cohorts.

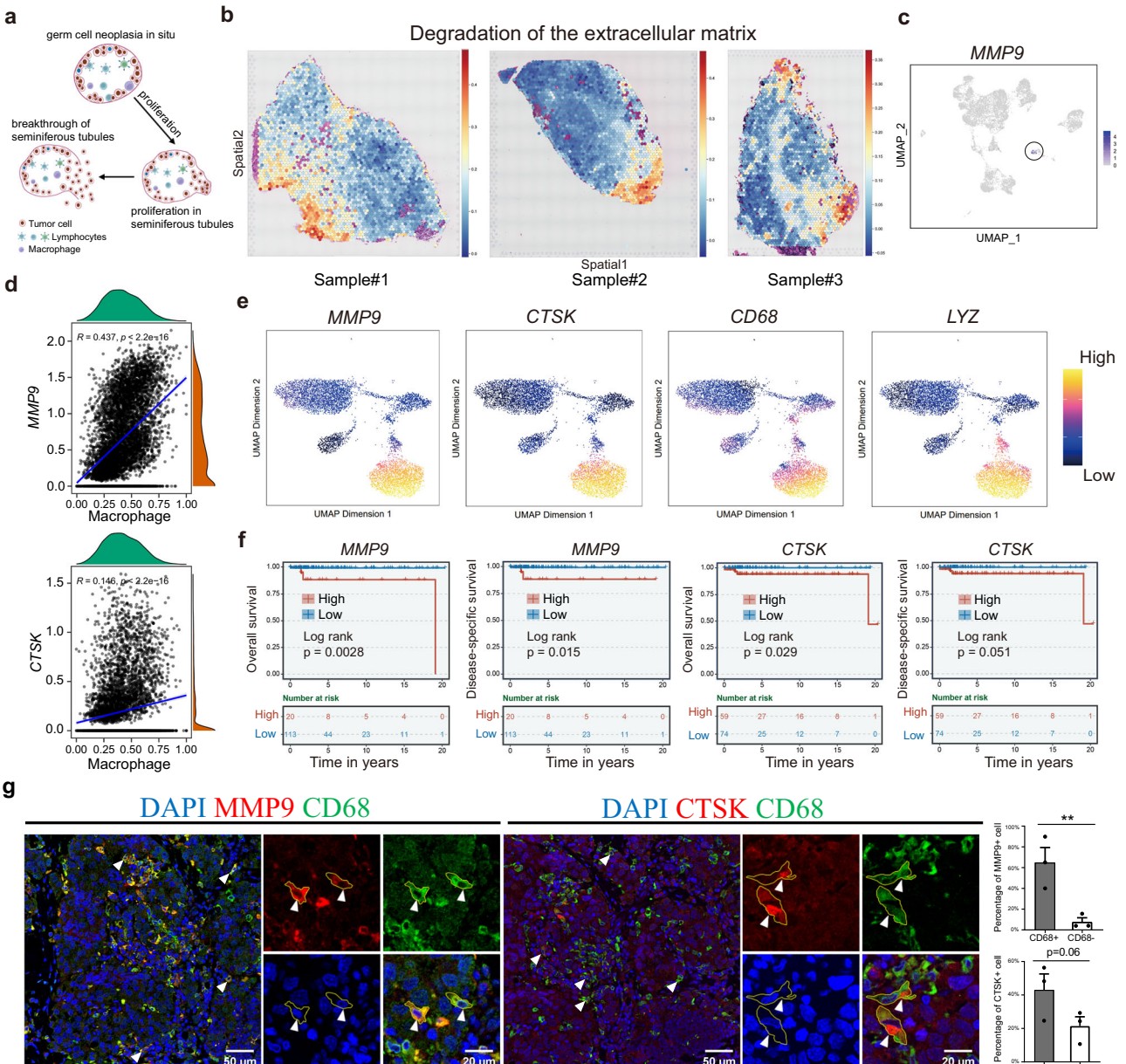

**Fig. 5 | *MMP9* and *CTSK* facilitated seminoma to break through the tubules.**
**a** Diagram of tumor cells breaking through the seminiferous tubules. This plot created with BioRender.com. **b** Expression of selected pathway in spatial transcriptome data of three seminoma samples. **c** scRNA-seq data showed that *MMP9* mainly expressed in macrophage. **d** Correlation analysis of *MMP9* and *CTSK* expression with macrophage location in spatial transcriptomics data using Pearson's correlation coefficients. Points represent the cells in macrophage. Exact pvalue of *MMP9* and macrophage: 2.08E−296; Exact pvalue of *CTSK* and macrophage: 5.01E−32. Source data are provided as a Source data file. **e** *MMP9* and *CTSK* expression patterns in immune cells of scATAC-seq data from Fig. 3b. *CD68* and *LYZ*: macrophage

markers. **f** Survival analysis showing that *MMP9 and CTSK* high expression was associated with worse overall survival (OS) and Disease-specific survival in the TCGA cohorts. **g** IF staining of MMP9 and CTSK in tissue sections of seminoma patients (*n* = 3). White arrow represents positive signal. The barplot showed the percentage of MMP9+/CTSK+ cells in CD68+/CD68− cells. The height of the bar represents the cell ratio of MMP9+/CTSK+ cells in CD68+/CD68- cells. *P* value was calculated by one tailed paired t test. **\*\*P < 0.01**. Data are presented as "mean values ± SEM" as appropriate. Source data are provided as a Source data file. Scale bar, 50 μm and 20 μm in magnified regions.

We also noticed *MIF* and its receptors genes expressed in other cell types, suggesting frequent crosstalk between the tumor and immune cells (Fig. 6c). We further analyzed the expression of genes in the MIF pathway using spatial data, and we found that MIF was mainly expressed within the tumor regions, while the receptors were predominantly expressed in non-tumor cells (Fig. 6d and Supplementary Fig. 6d). To further validate, we conducted immunostaining and detected MIF and CD74 signals at the protein level in the seminoma tissues. We observed MIF, a secreted protein, had a colocalization with CD74 (Fig. 6e). At the protein level, cells with high MIF

expression had a lower expression of CD74, which consistent with our findings in scRNA-seq data (Fig. 6c, e). To further understand the impact of MIF on immune cells, we separately examined the influence of MIF on macrophages and B cells. Macrophages in tumor microenvironment existed in two polarized states M1 and M2. M1 macrophages are considered to have anti-tumor properties, whereas M2 macrophages historically exhibit pro-tumorigenic functions[69,70]. By analyzing the enrichment of M1 and M2 signatures in macrophages, we observed a significant prevalence of M2 macrophages within seminoma microenvironment, indicating that macrophages in

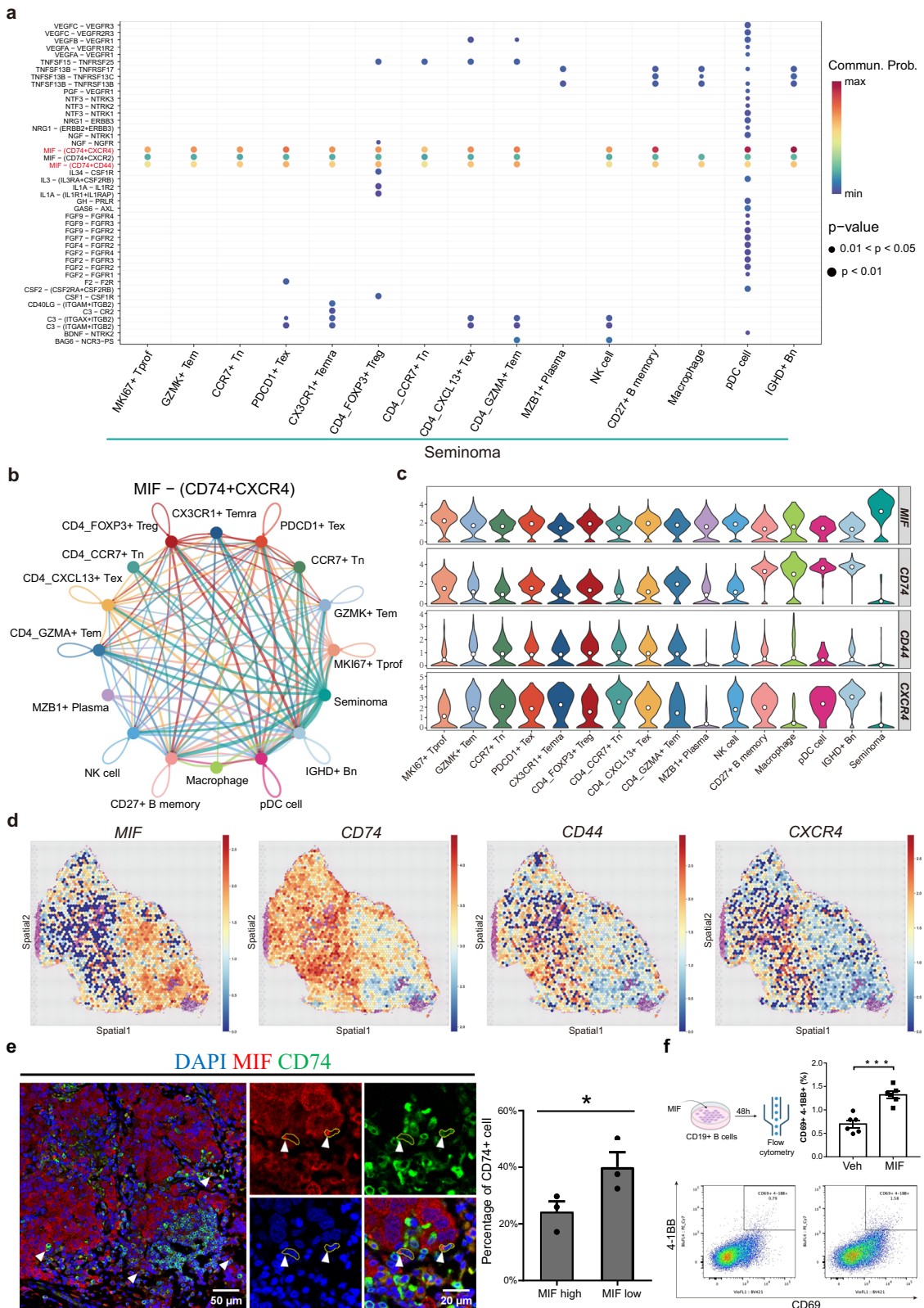

seminoma TME adopt an immunosuppressive state (Supplementary Fig. 6e). We also conducted in vitro culturing experiment by treating cultured B cells with MIF. Our results demonstrated a significant increase in the proportion of MIF-stimulated B cells exhibiting activation by expressing CD69 and CD137 (4-1BB). (Fig. 6f). These results suggested a potential tumor-promoting role of MIF in seminoma via communication with immune cells.

## Discussion

Although seminoma is among the most common malignancies in young adult men, it has not been extensively studied due to high curability. However, there are major long-term health consequences seen in seminoma patients post cancer treatment due to the non-specific treatments employed. To gain a deep understanding of seminoma tumorigenesis and TME, we conducted a series of extensive

**Fig. 6 | Tumor cells secreted MIF to affect other cells. a** Interacting ligand-receptor (L-R) pairs from seminoma to immune cells in TME. *P* value was calculated by Wilcoxon rank sum test. **b** The interaction network of all cells by MIF-(CD74 + CXCR4) L-R pairs in the microenvironment, with tumor cells being the main output signaling cells. The color of the line represents the signal emitted from the cell with the corresponding color. **c** Gene expression of MIF pathway in immune cell sub-types. **d** Spatial expression pattern of ligand (*MIF*) and receptor (*CD74, CD44* and *CXCR4*) in MIF pathway. **e** IF staining of MIF and CD74 in seminoma samples (*n* = 3). White arrow represents positive signal. The barplot indicated the percentage of CD74+ cells in cells with high (in tumor cells) and low (in immune cells) MIF expression. *P* value was calculated by one tailed paired t test. \**P* < 0.05. Data are presented as "mean values ± SEM" as appropriate. Source data are provided as a Source data file. Scale bar, 50 μm and 20 μm in magnified regions. **f** Proportion of activated B cells detected by flow cytometry after co-culture with MIF in vitro. Barplot showing the activated B cell ratio in control group (Veh) and MIF stimulated group (MIF) (*n* = 6). *P* value was calculated by two-sided Wilcoxon rank-sum test. \*\**P* < 0.01. Data are presented as "mean values ± SEM" as appropriate. Source data are provided as a Source data file. Cell culture schematic (upper left) created with BioRender.com.

experiments by leveraging cutting edge molecular and genomics techniques to assess seminoma patient samples. We found that seminoma and PGCs display strong similarities in their transcriptomes, especially by sharing the expression of key TFs such as TFAP2C, POU5F1, NANOG and SOX17. Previous studies only examined the expression patterns of a few key markers, while our work provided additional genes through unbiased transcriptomic profiling and analysis. Notably, previous studies propose that seminoma may originate from PGC-GCNIS, and our results reinforce this possibility.

TFAP2C is a key TF expressed in PGCs critical for germ cell lineage specification. Our work confirmed its high expression in seminoma, and, through single-cell chromatin accessibility analysis, showed its binding motif is highly enriched in seminoma, further promoting a possibility that it regulates gene expression in seminoma and has a functional impact. Indeed, through studies using a seminoma cell line TCam-2, we proved that *TFAP2C* knockdown can lead to decreased cell invasion and migration. Previous studies showed that TFAP2C can promote tumor progression and is associated with poor prognosis in breast and lung cancers[71–73]. Here, given its close relationship with PGCs, seminoma clearly has a much easier route to obtain or maintain *TFAP2C* expression, which may explain its high occurrence in young adult men. Interestingly, although sharing *TFAP2C* expression, seminoma is much less malignant than breast and lung cancers, suggesting that additional factors counteract the role of TFAP2C in seminoma.

Previous studies have reported that there is immune infiltration in seminoma, such as B cells, T cells, and DCs, while lacking functional description of them[27]. We identified 15 immune cell subtypes in the seminoma TME. The quantity and type of immune cells in seminoma is significantly more than that in the healthy normal testis, suggesting complex immune-tumor interactions. We found that in seminoma, a large proportion of immune cells are in a dysfunctional or exhausted state, while there is a significant part which have effector responses, indicating immune cells and tumor cells in a dynamic balance. However, based on the spatial distribution and proportion of tumors and immune cells, the seminoma TME seems to be in an immune-suppressed state, which may contribute to the progression of seminoma.

Additionally, we found that *MIF* is highly expressed in seminoma, and its receptor mainly expressed in the immune cells, indicating the tumor had frequent crosstalk to the immune cells in TME. Moreover, *MIF* also highly expressed in metastasized seminoma from a previously published data (Supplementary Fig. 6f)[6]. Therefore, we propose *MIF* as a promising candidate target to treat seminoma, though additional studies are needed to further investigate its functional impact in seminoma and the underlying mechanism.

## Methods
### Ethics statement
Seminoma samples used for scRNA-seq were obtained from four individuals, with written consent, through the University of Utah Andrology laboratory (IRB approved protocol #00075836). Studies with human specimens used for scATAC-seq and 10X Visium profiling were approved by the Ethics Committee of the Affiliated Cancer Hospital of Xiangya School of Medicine, Central South University (2021KYKS-46). All of these samples were in-house. Informed consent was obtained from each patient before surgery. The detail information for samples see Supplementary Data 1.

### Sample transportation and storage
Upon surgical extraction, the testes were promptly placed in containers on ice, where they were left for 1–2 h during transportation. Following transportation, the tunica was removed, and the testicular tissues were sliced into small pieces of approximately 500 mg to 1 g each. 90% of these tissues were expeditiously moved into cryo-vials (Corning) preloaded with 1.5 ml of freezing medium consisting of 75% DMEM medium (Life Technologies), 10% DMSO (Sigma-Aldrich cat #D8779), and 15% fetal bovine serum (FBS) (Gibco). The cryovials were thereafter placed in an isopropanol chamber (Thermo Fisher Scientific) and stored in a −80 °C refrigerator overnight. The following day, the cryovials were relocated to a liquid nitrogen tank, where they could be stored for an extended period.

### Sample Fixation for Immunostainings
Tissues were fixed in 4% paraformaldehyde (PFA) for 24 h, followed by thorough washing with 0.01 M PBS for 10 min x 3 times to remove the PFA. The fixed tissues were dehydrated using concentration gradient ethanol and gradually transitioned to xylene. Tissues were placed in 50% ethanol, 75% ethanol, 80% ethanol, 95% ethanol I, 95% ethanol II, 100% ethanol I, 100% ethanol II, 1/2 absolute ethanol and 1/2 xylene, xylene I, and xylene II for clear dehydration (the soaking time in each concentration liquid was about 30–50 min and could be adjusted according to room temperature). Finally, the tissues were embedded in paraffin.

### Immunohistochemical staining
The tissue embedded in paraffin was sliced, and the tissue sections were then placed onto slides and heated at 65.5 °C for 1 h. The slides were subsequently deparaffinized through a series of xylene washes (I for 10 min, II for 10 min, III for 10 min), followed by rehydration in a descending ethanol series (100% for 10 min, 90% for 5 min, 80% for 5 min), and water for 5 min. Antigen retrieval was performed followed by two washes in PBST for 5 min each. Then, each tissue section was incubated with 0.3% hydrogen peroxide enzyme blocking solution for 10 min, followed by two washes in PBST. The sections then underwent a second non-specific protein blocking step followed by two washes in PBST. After blockage, the sections were incubated with primary antibody included anti-CD45 (#20103-1-AP, Proteintech, 1:400), anti-VIM (#5741 T, Cell Signaling Technology, 1:300), anti-NANOG (#ab109250, Abcam, 1:100), anti-OCT4 (#sc-5279, Santa, 1:200), anti-CD3 (#MAB-0740, MXB Biotechnologies, 1:200), anti-PDCD1 (#P04417, ProMab Biotechnologies, 1:200), anti-TFAP2C (#14572-1-AP, Proteintech, 1:100) overnight (12–16 h) at 4 °C. Once washed, these were then incubated with secondary antibody for 90 min in darkness. Subsequently, DAB (3,3'Diaminobenzidine) was added while observing the color development under a microscope. Stain the tissue with hematoxylin and eosin (H&E) staining after visualization, then sequentially immerse the slide in 80%, 90%, 100%, xylene I, xylene II, and xylene III for 2 min each.

## Immunofluorescence staining

The paraffin sections were deparaffinized with xylene, respectively xylene 1, xylene 2, xylene and ethanol 1:1 mixture, 100% ethanol, 100% ethanol, 95% ethanol, 85% ethanol, 70% ethanol, 50% ethanol each for 5 min, followed by 5 min of PBS washing on a shaker. After that it was used for about 300 ml of 1×Tris-EDTA (PH 9.0, supplemented with Tween 20 to 0.05%) antigen repair solution, heated in a plastic dye box in a microwave oven on high for about 5 min, and then stopped heating after boiling, put the slices into the microwave oven, and repaired on low fire for 5 min, and then took out the slices and examined and opened the lid to dissipate the heat for a few seconds and then continued to be repaired on low fire for 10 min. When closing, use a histochemical pen to draw a circle slightly larger than the boundary of the tissue, and then add 60 µl of 5% BSA/PBS to the surface of the tissue, and close the wet box at room temperature for 30 min. Dilute the primary antibody to the corresponding concentration with the closure solution and then add 60 µl of the primary antibody included anti-SOX17 (#AF1924, R&D systems, 1:50), anti-CD19 (#ab134114, Abcam, 1:200), anti-CD68 (#66231-2-Ig, Proteintech, 1:3200), anti-MMP9 (#10375-2-AP, Proteintech, 1:400), anti-CTSK (#11239-1-AP, Proteintech, 1:400), anti-MIF (#20415-1-AP, Proteintech, 1:800), anti-CD74 (#66390-1-Ig, Proteintech, 1:1600) to the surface of the tissue, and hybridize in the wet box at 4 °C overnight. Remove and rewarm for half an hour, wash with PBST (0.1% Tween 20) on a shaking table for 3 × 5 min, then dilute the secondary antibody with PBST to the corresponding concentration and add 6 µl drops to the surface of the tissue, and hybridize in the wet box at room temperature for 1 h. Wash with PBST (0.1% Tween 20) on a shaking table for 3×5 min. After washing, the liquid on the slide surface was blotted out, and 15 µl of sealer (containing DAPI) was added to the surface of the tissue, coverslip was applied, and the slides were sealed with nail polish.

## Cell culture and small interfering RNA (siRNA)

The TCam-2 cells were gifted by Dr. Riko Kitazawa (Department of Diagnostic Pathology, Ehime University Hospital, Matsuyama, Japan). The mycoplasma testing was conducted using the MycoAlertTM PLUS Mycoplasma Detection Kit (Catalog #: LT07-710, Lonza Bioscience) according to the manufacturer's instructions. After testing negative, the TCam-2 cells were cultured in Dulbecco's Modified Eagle's Medium (DMEM; GIBCO, USA) supplemented with 10% FBS (GIBCO cat # 10099141 C), 100 U/ml penicillin, and 0.1 mg/ml streptomycin (GIBCO cat # 15140122). The cells were maintained in a 37 °C, 5% CO2 humidified cell culture incubator.

For cell preparation, cells in logarithmic growth phase were digested and seeded at an appropriate density into culture dishes or plates. Transfection was performed when the cells reached a fusion density of 65%–80%. To prepare the transfection mixture, control and experimental siRNA were separately dissolved in OPTI, following the instructions provided by Ribo siRNA and Invitrogen's Lip3000, respectively. Simultaneously, an appropriate amount of Lip3000 was dissolved in OPTI corresponding to the culture dishes or plates. The dissolved siRNA and Lip3000 were then mixed with OPTI and incubated at room temperature for 15 min. Finally, the siRNA and Lip3000 complex in OPTI was added to each culture dish or plate, followed by incubation in a 37 °C humidified incubator. Our TCam-2 cell from Dr. Riko Kitazawa (Department of Diagnostic Pathology, Ehime University Hospital, Matsuyama, Japan) as a kindly gift.

## RNA extraction and qRT-PCR

The total RNA was extracted using Trizol. The complete culture medium was removed via pipette suction from the culture dish or culture plate, and then the cells were gently washed 2–3 times with pre-cooled 1× PBS. An appropriate amount of Trizol was added to the culture dish or culture plate (1 ml Trizol/60 mm culture dish, 0.5 ml Trizol/one well of the six-well plate). The cells were scraped down and collected into a pre-cooled 1.5 ml RNase-free centrifuge tube using the large end of a 200 µl tip. They were then placed on ice for 15–20 min. After, pre-cooled chloroform (0.4 ml) was added to the centrifuge tube, which was vigorously shaken on a vortex mixer for 1 min and then placed on ice for 5 min. Next, the centrifuge tube was centrifuged in a low-temperature centrifuge at 12,000 rpm/min for 15 min. The upper aqueous phase was carefully transferred to another pre-cooled 1.5 ml RNase-free centrifuge tube (taking care not to suck the lower white precipitate). To each centrifuge tube, 1 ml isopropanol was added, mixed well, and then placed in a refrigerator at 4 °C for 30 min. The centrifuge tube was centrifuged at 12,000 rpm/min for 10 min, the supernatant was discarded, and 1 ml of 75% ethanol prepared with DEPC water was added to each centrifuge tube. The centrifuge tube was vortex mixed and then centrifuged at 12,000 rpm/min in a low-temperature centrifuge for 5 min. The supernatant was discarded, and the previous step was repeated once. The centrifuge tube was inserted into a float and placed upside down on a clean filter paper for 15–20 min to dry. Finally, 20-30 µl of DEPC water was added to dissolve the RNA. It was blown and mixed with a pipette, and stored at −80 °C for subsequent use. The Transcript or First Strand cDNA Synthesis Kit (ROCHE cat # 04897030001) was employed for reverse transcription PCR reaction with 1 µg of RNA being used.

Primer sequence: *TFAP2C*: F: 5′-TCAGTCCCTGGAAGATTGTCG-3′; R: 5′-CCAGTAACGAGGCATTTAAGCA-3′.

*ACTB* (β-ACTIN): F: 5′- TCACCAACTGGGACGACATG-3′; R: 5′- GTCACCGGAGTCCATCACGAT-3′.

*TFAP2C*-siRNA: SiRNA1: 5′-CCGAUAAUGUCAAGUACGA-3′; SiRNA2: 5′-ACACUGGAGUCGCCGAAUA-3′.

## Cell migration

Cells used for the Transwell migration experiment were digested and washed twice with 1xPBS, and then resuspend the cells in serum-free medium. An appropriate amount of cell suspension was taken for cell counting, and the cell concentration was adjusted to 100,000 cells/ml based on the cell count. Each well of the 24-well plate used for placing the Transwell chambers was filled with 800 µl of culture medium containing 15% FBS. The Transwell chambers were then carefully placed into the plate to prevent the formation of bubbles between the bottom of the Transwell chamber and the medium. Subsequently, 200 µl of the aforementioned cell suspension (with a concentration of 100,000 cells/ml) was added to the upper chamber of each Transwell chamber. The entire 24-well plate was incubated in a cell culture incubator for 48 h. The Transwell chambers were removed from the 24-well plate, and the chambers were rinsed twice with physiological saline. The cells on the chambers were fixed with formalin solution at room temperature for 30 min. Afterward, the chambers were rinsed three times with water. The chambers were placed in 0.1% crystal violet solution and left at room temperature for 10 min. Following staining, the chambers were rinsed with sufficient physiological saline to remove excess crystal violet staining solution. Finally, the cells that did not pass through the Transwell chamber on the surface of the Transwell chamber were gently removed using a sterile cotton swab. The cell staining was observed under an inverted microscope, and photos were taken to count the number of cells.

## Cell invasion

The BD Matrigel gel was thawed in advance at −20 °C and allowed to melt in the refrigerator at 4 °C overnight. The sterile tips and centrifuge tubes needed for BD Matrigel gel were pre-chilled in the −20 °C freezer for 2 h. Then the pre-chilled centrifuge tubes were retrieved and added 200 µL of serum-free culture medium to each tube. 50 µL of BD Matrigel gel was drawn up with a pipette and added to the tubes, and then gently mixed the contents with the pipette. After, the diluted BD Matrigel gel was added to each upper chamber of the Transwell plates, with 30 µL added to each chamber. The prepared chambers

were placed in a 24-well plate and incubated at 37 °C for 4 h to allow the Matrigel gel to solidify. After preparation, the TCam-2 cells were digested from dishes and washed twice with 1×PBS. The cells were resuspended in serum-free medium and the cell concentration was adjusted to 100,000 cells/mL. 800 μL of medium containing 15% fetal bovine serum was added to each well of the 24-well plate for Transwell chambers. The Transwell chambers were carefully placed into the wells, ensuring no formation of bubbles between the chamber bottom and the medium. Next, 200 μL of the above cell suspension with a concentration of 100,000 cells/mL was added to each upper chamber of the Transwell. The entire 24-well plate was incubated in a cell culture incubator for 48 h. After the incubation period, the Transwell chambers were removed from the 24-well plate, washed twice with physiological saline, and then the cells on the chambers were fixed with formalin solution at room temperature for 30 min. The chambers were rinsed three times with water. Subsequently, the chambers were placed in a 0.1% crystal violet solution at room temperature for 10 min. After staining, the chambers were washed with abundant physiological saline to remove excess crystal violet solution. Finally, the Matrigel gel on the surface of the Transwell chambers and the cells that did not pass through the chambers were gently removed using a sterilized cotton swab. The cell staining was observed under an inverted microscope, photographs were taken, and the number of cells was counted.

### B cells isolation and co-culture with MIF

Cells examined in this study were derived from peripheral blood mononuclear cells (PBMCs) purified from peripheral blood by density gradient centrifugation with Lymphoprep (STEMCELL Technologies). Total CD19 + B cells were prepared with MojoSort™ Human Pan B Cell Isolation Kit (Catalog:480082, Biolegend). B cells were cultured in full RPMI 1640 medium in the presence of 10 ng/mL recombinant human MIF protein (Catalog: 300-69-5, peprotech) for 48 h. In the last 24 h, the culture medium was supplemented with 100 ng/mL LPS (Sigma-Aldrich) as previously described[74–76].

### Flow cytometry

Cell surface molecule staining was analyzed as previously reported using the flow cytometer (Cytec, USA)[74,77]. Zombie Aqua fixable viability kit (catalog: 423101, 1:200), APC-anti-human CD19 (catalog: 302212, 1:100), PE-Cy7-anti-human 4-1BB (catalog: 309817, 1:100) were purchased from Biolegend (USA). Brilliant Violet 421-anti-human CD69 (catalog: 562884, 1:100) was purchased from BD(USA). Data analyses were carried out with FlowJo 10.0 software.

### Sample preparation for single cell RNA sequencing

For each single-cell RNA sequencing experiment, a single cryovial of tissue was swiftly thawed. The tissues were rinsed twice with PBS and then scraped with razor blades, and then enzymatically digested using 1 mg/ml collagenase type IV (Sigma Aldrich)/1 mg/ml DNase I (Sigma Aldrich) followed by the digestion with trypsin-EDTA (Invitrogen)/1 mg/ml DNase I at 37 °C for 5 min. Afterward, single cells were filtered through 40 μm strainers (Thermo Fisher Scientific) and washed with D-PBS (Thermo Fisher Scientific). At last, the cells were resuspended in D-PBS + 0.4% BSA (Thermo Fisher Scientific) at a concentration of ~1000 cells/ml, rendering them ready for single-cell sequencing.

### Single cell RNA-seq performance, library preparation and sequencing

The protocol used for the experiment was based on the user guide provided by 10X Genomics for Chromium Next GEM Single Cell 3′ Reagent Kits v3.1. To generate and barcode the GEMs, cells were diluted to achieve approximately 5000 cells per lane and were loaded with the master mix onto the Chromium Next GEM Chip G. Following post-GEM-RT cleanup, cDNA Amplification was conducted using 12 cycles. The resulting libraries were then sequenced on an Illumina

Novaseq instrument using the following settings: 28 cycles for Read 1, 10 cycles for i5 index, 10 cycles for i7 index, and 90 cycles for Read 2.

### CUT&Tag library generation

CUT&Tag was performed with Hyperactive Universal CUT&Tag Assay Kit for Illumina (Vazyme Biotech, # TD903) following the manufacturer's instructions. In brief, $1 \times 10^5$ TCam-2 cells were collected and washed with 500 μl Wash Buffer. Cells were resuspended with 100 μl Wash Buffer after centrifugation (600 × g, 5 min, RT). 10 μl ConA Beads (Pre-washed twice with Binding Buffer and resuspended in 10 μl Binding Buffer) were added to the cell tube and incubated at room temperature for 10 min. After discarding the supernatant, the cell - magnetic bead complexes were resuspended with 50 μl pre-cooled Antibody Buffer containing 1ul anti-TFAP2C antibody (Santa Cruz Biotechnology, #sc-12762) or normal mouse IgG (Santa Cruz Biotechnology, #sc-2025) and rotated at 4 °C overnight. The supernatant was discarded, and 50 μl Dig-wash Buffer with Goat anti-mouse IgG (1:100) was added and incubated at room temperature for 1 h the next day. Then the cell - magnetic bead complexes were washed gently with 200 μl Dig-wash Buffer three times, and incubated with 0.04 uM pA/G-Tnp (2 μl pA/G-TnP with 98 μl Dig-300 Buffer) at room temperature for 1 h. In the same way, the cell - magnetic bead complexes were washed three times with 200 μl Dig-300 Buffer. Added 50 μl tagmentation buffer containing 10 μl 5 x TTBL and 40 μl Dig-300 Buffer and incubated at 37 °C 1 h. The interaction was stopped with 5 μl 20 mg/ml Proteinase K, 100 μl Buffer L/B and 20 μl DNA Extract Beads at 55 °C for 10 min. The supernatant was discarded, and the complexes were washed once with 200 μl Buffer WA then 200 μl Buffer WB twice, resuspended with 22 μl nuclease free water to dissolve the DNA. PCR was performed to amplify the libraries with 15 μl purified DNA, 25 μl 2 x CAM, 5 μl barcoded i5 and i7 primers from "TD202 TruePrep Index Kit V2 for Illumina" (Vazyme, # TD202), and using the following program: 72 °C for 3 min, 98 °C for 3 min, 15 cycles of 98 °C for 10 s, 60 °C for 5 s, and 72 °C for 1 min, and holding at 4 °C. To purify the PCR products, added 2x volumes of VATHS DNA Clean Beads (Vazyme Biotech, # N411) and incubated at room temperature for 5 min. The supernatant was discarded and the beads were washed twice with 200 μl fresh 80% ethanol and eluted in 22 μl nuclease free water. CUT&Tag libraries were sequenced by Anoroda using the Illumina NovaSeq 6000 platform in PE150 mode (Anoroda, Beijing, China).

### Single-cell ATAC−seq library generation

The collected tissues were washed with 1X PBS, quickly frozen, and subsequently stored in liquid nitrogen. Nuclei extraction was performed using the mechanical extraction method[78]. Firstly, collected tissues were added to a 2 ml Dounce homogenizer set and thawed in homogenization buffer (containing 20 mM Tris pH 8.0, 500 mM sucrose, 0.1% NP-40, 0.2U/μl RNase inhibitor, 1X protease inhibitor cocktail, 1% bovine serum albumin (BSA), and 0.1 mM DTT). Tissues were manually homogenized using Dounce pestle A ten times, filtered through a 70 μm cell filter, followed by ten additional homogenization steps with Dounce pestle B, and filtering through a 30 μm cell filter. Nuclei were pelleted by centrifugation at 500 × g for 5 min at 4 °C, resuspending in blocking buffer containing 1% BSA and 0.2U/μl RNase inhibitor in 1X PBS. After centrifugation at 500 × g for 5 min, nuclei were resuspended with Cell Resuspension Buffer (MGI).

Single-nucleus ATAC-seq libraries were generated using the DNBelab C Series Single-Cell ATAC Library Prep Set (MGI, #1000021878)[79]. In brief, transposed single-nucleus suspensions were converted to barcoded scATAC-seq libraries through the following steps: droplet encapsulation, pre-amplification, emulsion breakage, capture beads collection, DNA amplification, and purification. Indexed sequencing libraries were prepared according to the user guide, with the concentrations of sequencing libraries measured using the Qubit ssDNA Assay Kit (Thermo Fisher Scientific). All libraries underwent

paired-end 50 sequencing using the ultra-high-throughput DIPSEQ T1 sequencer platform.

### 10X Visium library generation

Before sequencing, we assessed RNA quality using DV200, DV200 ≥ 50% be used. We then conducted a tissue adhesion test, followed by deparaffinization and H&E staining. After staining, we captured images of the area using brightfield settings and released the RNA through decrosslinking. We then added an integrative transcriptome probe panel, which consisted of a pair of specific probes for each target gene, to the previously stained tissue. Probe pairs hybridized to their complementary target RNA and formed a stable ligation product with the addition of a ligase. After processing the ligation structure, we released the single-stranded ligation products from the tissue. RT Master Mix containing reverse transcription reagents was then added for amplification, followed by the addition of UMI, barcode and partial Read1. The library with barcoded ligation products was released to qPCR to determine cycle amplification number, added with index, and finally constructed the library containing P5 and P7 primers used for Illumina amplification. The library molecules were cleaned up by SPRIselect and assessed, quantified, and sequenced on a bioanalyzer or similar instrument using PE150 mode sequencing on the Illumina Nova-Seq600 platform.

### Quantifications and statistical analysis

**Processing of single cell RNA-seq data.** The raw data was demultiplexed using the *mkfastq* function of Cell Ranger v7.0.0. The resulting Fastq files were then processed through the *count* function using default settings, which involved alignment (using STAR align with *GRCh38* human reference genomes), filtering, and UMI counting.

Use the *Read10X* function in *Seurat* (https://satijalab.org/seurat/index.html, R package, v4.2.0) to sequentially load UMI count matrices generated from four samples into R. After adding sample information to the row names of the matrix, all of this information was merged to create a Seurat object using *merge* function. According to the developer's description, the data has undergone filtering and normalization. Cells with >500 expressed genes were retained, and <25% of reads mapped to the mitochondrial genome. UMAP and clustering analysis were performed on the merged dataset using the top 2000 highly variable genes and 1-30 PCs. Identification of cell types performed based on the tutorial. When analyzing individual cell subpopulations, the *subset* function was used to extract cells based on cell type annotations and form new Seurat objects. The Seurat objects of CD8 + T cells and CD4 + T cells were converted into CellDataSet objects for importing into Monocle package (v2.26.0). Performed pseudotime analysis according to the default settings.

**scATAC-seq data analysis.** The data from scATAC-seq was analyzed using the R package *ArchR* (https://github.com/GreenleafLab/ArchR, v1.0.2).

After creating Arrow files, we utilized the *ArchRProject* function to create an ArchR file, which integrates seminoma and healthy testicular samples. Following the developer's tutorial, we used the *addIterativeLSI* function to select 1-30 PCs for dimensionality reduction and clustering, and calculated gene scores using default functions. Based on the promoter accessibility and different cell types marker gene activity scores, we manually identified the corresponding cell types for each cluster.

We then used the *subsetArchRProject* function to extract tumor cells and germ cells and reclassified them. ScRNA-seq data and scATAC-seq data were integrated using the *addGeneIntegrationMatrix* function for accurate annotation of the cell types. *AddGroupCoverage* function were used to pool together all single cells. MACS3 (v2.2.7.1) was used for peak calling to identify specific peaks for each cell type. Using the *addArchRGenome* function to set the default reference genome as hg38. Finally, we enriched motifs and identified the specific binding transcription factors for each cell type using the *addMotifAnnotations* function.

**CUT&Tag-seq analysis.** After download the raw data, we conducted quality control using FastQC (v0.11.9), and eliminated low-quality bases using trim_galore (v0.6.7). Paired-end reads were aligned to the GRCh38 reference genome using bowtie2 (v2.2.5) with options: --local --very-sensitive --no-mixed --no-discordant --phred33 -I 10 -X 700. Unmapped and low mapping quality reads were filtered using samtools (v.1.6), and duplicates were removed using picard MarDuplicates (v3.0.0). The bam files were converted to BigWig files using deeptools bamCoverage (v3.5.1). Peaks were called using MACS2 (v2.2.7.1) with the following parameters: -t input.bam -c control.bam -p 1e-5 -g hs -f BAMPE --keep-dup all. Annotation of peaks was performed using ChIPseeker (v1.36.0).

**Spatial transcriptomes analysis.** The Visium spatial transcriptomics array was processed using the cprovided by 10X Genomics, which enabled the alignment and summarization of unique molecular identifier (UMI) counts against the hg38 human reference genome for every spot.

The spatial transcriptomics data matrix were analyzed by the python package *Scanpy* (https://github.com/scverse/scanpy-tutorials, v1.9.1). Data for each sample was read in using the *sc.read_visium* function, and quality control and filtering were conducted as per the developer's guide. 3000 highly variable genes were selected using the *sc.pp.highly_variable_genes* function for normalization and scaling. PCA and clustering were performed using 1-50 PCs, and the cell clusters were visualized on HE-stained spatially-mapped images using the *sc.pl.spatial* function. *Sc.pl.spatial* was used to visualize marker genes expression and distribution of each cluster. The Scanorama algorithm was used to integrate the single-cell RNA-sequencing and spatial data and annotate cell types[80]. The *sc.tl.score_genes* function was used to calculate enrichment scores for genes related to specific pathways and map them to spatial locations.

**Cell-cell communication analysis.** Cell-cell communication used R package CellChat (https://github.com/sqjin/CellChat, v1.4.0) to analyze. Based on the cell information and RNA expression from scRNA-seq data, we first create a CellChat object using *createCellChat* function. For downstream analysis, we utilized ligand-receptor interactions database, including "Secreted Signaling," "ECM-Receptor," and "Cell-Cell Contact" databases. We used functions provided in the developer's tutorial to project gene expression data onto a protein-protein interaction network (PPI). The *computeCommunProb* function was utilized to calculate the communication probability between cells and infer the interaction network. Additionally, we used the *computeCommunProbPathway* function to infer intercellular interactions at the signal pathway level. For this, we made tumor cells as *source.use* and other cell types as *target.use* to determine the impact of signals emitted from tumor cells on other cells in their microenvironment.

**Survival analysis and pathway enrichment analysis.** The survival curve analysis was plotted by BEST online tool (https://rookieutopia.com/app_direct/BEST/) based on the TCGA TGCT cohort analysis. And all GO terms enrichment analyses mentioned in the article were conducted using the online website DAVID (https://david.ncifcrf.gov/tools.jsp). Terms with p-value < 0.05 were considered significantly enriched. The enrichment score of M1 and M2 were calculated in macrophage using the function *AddModuleScore* in Seurat package at single-cell level. The enrichment score of "Degradation of the extracellular matrix" was performed by functions *sc.tl.score_genes* in Scanpy package. The detailed gene list show in Supplementary Data 2.

**InferCNV analysis.** To further identify malignant cells, we used inferCNV (https://github.com/broadinstitute/inferCNV) to find evidence of somatic alterations in individual cells with large-scale chromosome copy number variants. We extracted seminoma cells from the Seurat object, and then using T cells, which do not normally undergo copy number variation, as reference cells for inferCNV analysis. We preformed inferCNV analysis using the default parameters.

**Principal component analysis (PCA).** The R package plotly (https://github.com/plotly/plotly.R, v4.10.2.9000) was used to visualize 3D PCA in Fig. 2d. For PCA of different cell types within each sample shown in Supplementary Fig. 1c, we first calculated the average expression levels of the top 100 marker genes for each cell type in each sample using the function *AverageExpression* in Seurat package, which is a pseudo-bulk process. Then, we analyzed each cell type of each sample as a separate dataset and performed PCA using the *prcomp* function in the stats package (v4.2.3) in R. Finally, we use the package ggplot2 to generate a plot, where different colors and shapes were used to represent cell types and samples, respectively. The detailed data in Supplementary Data 3.

### Reporting summary
Further information on research design is available in the Nature Portfolio Reporting Summary linked to this article.

## Data availability
The raw sequencing data generated in this study have been deposited in the Genome Sequence Archive at National Genomics Data Center, China National Center for Bioinformation/Beijing Institute of Genomics, Chinese Academy of Sciences (https://ngdc.cncb.ac.cn/gsa-human, accession no. HRA004404 for scRNA-seq data (https://bigd.big.ac.cn/gsa-human/browse/HRA004404); accession no. HRA004502 for scATAC-seq data (https://bigd.big.ac.cn/gsa-human/browse/HRA004502); accession no. HRA004398 for 10X Visium spatial transcriptome data (https://bigd.big.ac.cn/gsa-human/browse/HRA004398) and accession no. HRA006112 for CUT&Tag data (https://bigd.big.ac.cn/gsa-human/browse/HRA006112)) that are publicly accessible. The processed expression matrices in this paper have been deposited in the OMIX, China National Center for Bioinformation / Beijing Institute of Genomics, Chinese Academy of Sciences (https://ngdc.cncb.ac.cn/omix, accession no. OMIX004217). The publicly available data of normal germ cell scRNA-seq and infant data included in early germ cells used in this study are available in the GEO database under accession code GSE120508[34]. The publicly available data of early germ cell scRNA-seq used in this study are available in the GEO database under accession code GSE143356[12]. The remaining data are available within the Article, Supplementary Information, Source Data file. Source data are provided with this paper.

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

## Acknowledgements

This work was supported by the National Key Research & Developmental Program of China (2022YFC2702600) to J.G. and Z.Z., and China Postdoctoral Science Foundation (2022M713135) to X.W. This work was supported by grants from the Natural Science Foundation of Hunan Province (2021JJ41091). This work was supported by the Wisdom Accumulation and Talent Cultivation Project of the Third Xiangya Hospital of Central South University (YX202108). We thank Dr. Riko Kitazawa (Department of Diagnostic Pathology, Ehime University Hospital, Matsuyama, Japan) for kindly gifted TCam-2 cell line. We also thank Yifan Bao for assistance in drawing schematic.

## Author contributions

J.G., H.B. and Z.Z. conceived and supervised the project. X.N., Y.L. and J.M.H. collected and processed tumor samples. X.L. conducted computational analyses with assistance from B.Z. and X.W.; L.R., G.M., Q.Z., Z.L. and L.F. performed validation experiments; the manuscript was written by J.G. and X.L. and agreement of all authors.

## Competing interests

The authors declare no competing interests.
