## [Peer Review File · Nature Communications]

Single-cell Multi-omics Analysis of Human Testicular Germ Cell Tumor Reveals its Molecular Features and MicroenvironmentReviewers' Comments:

Reviewer #1:

Remarks to the Author:

Lu and coworkers performed an interesting study based on multi-omics (specifically single cell and special transcriptomics) of human testicular germ cell tumors, especially seminoma as well as seminoma cell line. Confirmatory data were identified related to published results, and various immune subtypes were described (n=15). In addition, insights related to invasive growth was found.

Although of interest, there are a number of limitations of the study. The main ones are highlighted here:

- It is not clear why seminoma has been selected for this study. From a clinical point of view this is the least relevant one, for which effective treatment protocols are available (when surgery is not sufficient);
- The finding of the developmental stage of seminoma (especially related to PGC) is nice to see from this multi-omics approach as well, but has no novel aspects;
- Quite a number of techniques are used (like ATAC-seq to validate scRNAseq) although the additional value of those are not clear, and again the well known targets are identified;
- The study is becoming less straightforward from Figure 3 onwards. The CUT & TAG results indicate that 7210 genes are bound and regulated by TFAP2C (that is a lot);
- The representative nature of the TCam-2 cell line for seminoma is the best so far, but not absolute as reported before. In that context, performing one cell line for the KO approaches is quite thin in making a firm conclusion. Therefore the statement "Altogether, these results revealed the key role of TFAP2C in promoting migration and invasion in seminoma by directly regulating genes such as FOSL1, CCN3 and LYPD3" is premature. In addition, the state of development to be investigated in this set up will be the transition between GCNIS and seminoma;
- Related to the immune profiles, the numbers are quite limited. In addition Figure 4 is irrelevant based on the clinical behavior of seminoma (see introduction);
- Figure 5 and 6 as well as the related text is an overinterpretation of the data. This needs to be adjusted to be realistic related to the results shown. In addition, more validation will be needed.

Overall, the manuscript includes a number of interesting aspects, but besides the confirmatory data (which are well described), the more novel results are still preliminary.

Reviewer #2:

Remarks to the Author:

Summary:

Lu et al. performed single cell RNA-seq and ATAC-seq to characterize gene regulatory programs in various cell types from human testicular germ cell tumor samples. From functional perturbation assay, their data revealed the roles of TFAP2C in seminoma contributing to the growth, migration, and invasion programs. Furthermore, the authors identified the distinct distribution of immune cells in the seminoma microenvironment through spatial transcriptome profiling. Their results indicated that exhausted immune cells are enriched within tumor regions while effector immune cells are mainly located in the stroma. From further analysis, the authors hypothesized that tumor-derived MIF might repress immune cell functions in the seminoma microenvironment. Overall, the authors compile a multi-omics atlas dataset for testicular germ cell tumors, which would advance molecular understanding during seminoma tumorigenesis.

Major comments:

1. The major goal of this study is to characterize gene regulatory programs in various cell types from

testicular germ cell tumor samples. After the authors integrated scRNA-seq data from four individuals with the scATAC-seq data generated from other human specimens, they only obtained 8 refined cell clusters containing both scRNA-seq and scATAC-seq data. However, immune cells are not included in those 8 refined cell clusters. As immune cells are the major population (>50%) within tumor tissue and play important roles during tumorigenesis, generating high quality multiome data for this diverse major population will be essential.

2. To help assess the data quality, several key experiments/validations are necessary:

- a) Tn5 transposase-based assay is prone to open chromatin bias. The authors need to provide more evidence to support the specificity of CUT&Tag data generated in this manuscript.
- b) Utilize orthogonal methods (immunohistochemistry or immunofluorescence staining) to verify sequencing-based results.
- c) Verify functional interaction between tumor-derived MIF signaling and immune cells.

3. The lack of quantitative comparisons and statistical methods in several figures:

- a) What kind of correlation analysis is used in Fig.2D, S2C?
- b) What is the scale for the heatmap shown in Fig.2E-G?
- c) The expression of those key marker genes is only present in the subtype immune cells. Macrophage is also a small fraction within the tumor. Therefore, it is not appropriate to use bulk RNA-seq data and correlate it with survival data as the p-value needs to be more significant in Fig. 4H, 5F.
- d) No quantification results in Fig. 3J, 5G, 6E, S3G.

Minor comments:

1. Please provide a scale bar and low-magnification images to illustrate the distribution of those markers within tumor sections in Fig. 1D.
2. Mislabeled GO terms in Fig. 3H; the red box should represent "Upregulated GO terms in siTFAP2C".
3. Include N in survival analysis Fig.4H.
4. MMP9 and CTSK don't seem colocalized in Fig.5D.
5. Please provide a scale bar in Fig. 5G, 6E.
6. Please provide more information on how CNV information is generated from single cell RNA-seq data in Fig. S1D?
7. Lack of group information in Fig. S3H.
8. Data presentation should be consistent. For instance, tissue sample 1 & 3 were used to illustrate the location of immune cells in Fig. 4F. But tissue sample 2 was used in Fig. 5D. In addition, two samples were shown in Fig. S4E but three samples were shown in S4F.

Reviewer #3:

Remarks to the Author:

In their research, Lu and colleagues conducted a comprehensive exploration of molecular characteristics and microenvironmental aspects using single-cell RNA-seq and ATAC-seq and spatial transcriptomics within seminoma patient specimens and cell lines. Through genomic profiling and functional comparative studies between seminoma cells and primordial germ cells, the team discovered critical gene expression programs, prominently featuring the gene TFAP2C, as well as 15 distinct immune cell subtypes in the tumor microenvironment (TME), each with unique spatial localizations. Overall, this study significantly extends our understanding of seminoma pathogenesis and illuminates potential therapeutic targets by establishing a multi-omics atlas. While the data fundamentally corroborates the propositions presented, there are several aspects that could benefit from further refinement and improvement. More specifically:

Major

1. The authors claimed that they provided the atlas for the seminoma, yet their study only

encompasses four samples. This limited sample size may hinder the universality of the findings. To enhance data reliability, the reviewer advises the researchers to expand their sample size.

2. The "Immune Cell Infiltration in Seminoma" section aims to further break down the immune cell clusters in Figure 1B. This section heavily focuses on the extensive analysis of T cells, providing little breakdown of the B cells and their role in the seminoma microenvironment. This section could be further improved by incorporating more analysis on the functionality of B cell subtypes (marginal zone, memory, naïve B cells) in the TME.

3. In Figure 2, the correlations between PGCs and seminoma are discussed, however there is some positive correlation. There is some similarity between the fetal SSCs state 0 and seminoma in the PCA analysis (2C) and the heatmap in 2D that is not discussed. This relationship should also be considered and discussed. Additionally, the Figure 2F Adult SSC states are not very informative as it is depicted. Since the authors are focusing on the distinctions between seminoma and PGC's, the adult SSCs could be removed from this figure.

4. In Figure 2E, the authors point out the shared specific expression of genes between seminoma and PGCs (TFAP2C, POU5F1, NANOG and SOX17). Figure 2E also has a very interesting cluster 3 with some genes that are upregulated in PGCs yet downregulated in seminoma. Similarly, cluster 1 has some genes that are enriched in seminoma but not PGCs. Since this section focuses on the analysis comparison between seminoma and PGCs, this would be important to discuss and enhance the analysis of this data.

5. Figure S2A and Figure 2E, it is not clear the numbering system of the clusters. Figure 2E has 5 clusters, but there are more than 5 clusters shown in Figure S2A.

6. In the text, the authors describe the use of spatial transcriptomics profiling for this dataset – suggesting that TFAP2C and SOX17 are highly expressed within areas of seminoma in the spatial data. It is not clear how the author defines the areas of seminoma as Figure 2G and S2 G using just TFAP2C and SOX17, when before this, the author mentions marker genes for seminoma as POU5F1+ 13, TFAP2C+ and NANOG+. To keep things consistent, it would be useful to have the spatial regions containing seminoma defined by these markers.

7. The healthy tissue and tumor tissue have very distinct clusters in the Figure 3A, does it mean there is very little cell type overlap between normal tissue and the seminoma tissue? Further, there is no labeling of the colors depicted in these clusters, which are necessary for the reader to interpret this data. According to the markers and corresponding legend in 3C: TFAP2C, POU5F1 and SOX17 are markers of seminoma, yet this seems to only apply to the top left cluster. Can the author clarify why the tumor clusters in 3A are inclusive of more than just that top left cluster? The authors should also annotate the clusters in Figure 3B, the way it is at present is uninformative.

8. In addition, Figure 5F appears to be somewhat ambiguous and perplexing in its current presentation. It is unclear from the visual data representation which curve corresponds to gene expression and which one signifies the survival rate. To improve readability and comprehension, it would be beneficial to delineate these two components more explicitly, perhaps through distinct labeling, color coding, or other methods of differentiation. This would ensure the accurate interpretation of the figure and its corresponding data.

Minor

1. In Figure 6E, are the IF images adjacent to those in 6D? If so, please depict the ROI in which the IF was done. If not, this data would be more cohesive if the IF staining was done on a section adjacent to the tissues that the spatial transcriptomics analysis was done on.

2. Figure S2F contains informative knowledge of the spatial clustering of the tumor tissue sections. The paragraph corresponding to this section mentions the gene expression profiles of each cluster. Since this is spatial data where visualization of clusters is important, I would suggest visualizing this data in Figure 2. It would also be useful to have annotations of the clusters on Figure S2F.

3. The authors describe cytotoxicity-related genes twice on page 7 (line 46) and page 8 (line 29). It would be useful to explain the selection of these genes in both instances.

4. Figure 3J depicts the multiple modalities that are highlighted in this study and thus should have a more extensive analysis.

5. There are also some minor issues regarding the content that could be modified. For instance, in the CUT&Tag library generation section under the methods category, the cell numbers were incorrectly recorded as 1×10^5 , rather than the correct form, 1×10^5 .

Authors' Response to Reviewers for NCOMMS-23-15034-T by Lu et al.

We thank all the reviewers for their thorough and constructive comments, which we have utilized to revise the manuscript. We provide a point-by-point response to all reviewer comments below, but first provide a summary of the main additions during revision:

First, we conducted further detailed analysis of the immune cells, including a combined analysis of scRNA-seq and scATAC-seq data to explore the key transcription factors (TFs) involved in regulating different immune cell types (Fig S4B), and focused analysis to decipher the functions of B cell subtypes (Fig S4D-E). Next, to ensure the quality and robustness of the CUT&Tag results, we re-did the CUT&Tag profiling with improved controls (IgG) and adopted more strict cutoffs, providing a more refined and rigorous gene target list (Fig 3J). We also performed additional IHC/IF staining experiments of key markers to better support our genomics findings using seminoma samples from additional patients, including TFAP2C (Fig 3E), CD3 & PDCD1 (Fig 4F), CD68/MMP9 & CD68/CTSK (Fig 5G), and MIF & CD74 (Fig 6E). Additionally, we conducted an in vitro experiment of B cells to investigate the potential functional interaction between MIF signaling and immune cells (Fig 6F). We also made clarification, modification and correction in the text and figures to present our results in a more clear and proper manner. All the changes in the manuscript are colored in red.

Here, we provide point-by-point responses to all reviewer comments, below:

REVIEWER COMMENTS

Reviewer #1, expertise in testicular germ cell tumour (Remarks to the Author):

Lu and coworkers performed an interesting study based on multi-omics (specifically single cell and spatial transcriptomics) of human testicular germ cell tumors, especially seminoma as well as seminoma cell line. Confirmatory data were identified related to published results, and various immune subtypes were described (n=15). In addition, insights related to invasive growth was found.

Although of interest, there are a number of limitations of the study. The main ones are highlighted here:

- It is not clear why seminoma has been selected for this study. From a clinical point of view this is the least relevant one, for which effective treatment protocols are available (when surgery is not sufficient);

Thank you for the comment. While we acknowledge that seminoma is indeed highly curable, we want to highlight the importance of studying this unique cancer type due to several reasons.

From a clinical perspective, although traditional cancer treatments are highly effective

in treating seminoma, a significant proportion of patients (about 15-30%) may experience relapse after undergoing first-line chemotherapy. Additionally, a considerable number of seminoma patients may face fertility issues, with some becoming azoospermic and others experiencing permanent hypogonadism due to the gonadotoxic effects of the cancer treatment, which affects both tumor cells and normal germline cells.

In the realm of basic research, the origin of seminoma is believed to be developmentally blocked primordial germ cells (PGCs) or gonocytes, as indicated by previous epidemiological and histological studies. However, the specific molecular mechanisms underlying this phenomenon remain largely unexplored. Seminoma, therefore, presents an ideal research model to delve into the molecular programs within early human male germline development, providing valuable insights into the gene expression network that leads to tumorigenesis.

In conclusion, the investigation of seminoma holds the potential to offer more targeted therapeutic approaches to reduce relapse rates and minimize gonadotoxicity. Moreover, by studying seminoma, we can gain a deeper understanding of human male germline development. We have outlined these rationales in the introduction section of our manuscript.

- The finding a the developmental stage of seminoma (especially related to PGC) is nice to see from this multi-omics approach as well, but has no novel aspects;

Thank you for your comment. We acknowledge the valuable contributions of previous studies in shedding light on seminoma through epidemiological and histological research, particularly in identifying key bio-markers for diagnostic purposes. In our research, we aimed to complement these existing findings by adopting a multi-omics approach to comprehensively investigate the molecular features of seminoma in a more systematic manner. Our study indeed confirms many well-established aspects of seminoma, including the expression of several markers commonly used for its diagnosis, such as TFAP2C and POU5F1/OCT4. However, we are excited to emphasize that our work goes beyond the current knowledge by making significant advancements in understanding seminoma.

Firstly, we have identified and provided several novel markers specific to seminoma, which could potentially be used for more precise and accurate diagnosis in the future (Fig 2E & S2D). These markers hold promise for enhancing diagnostic capabilities and improving patient outcomes. Secondly, our comprehensive analysis allowed us to delve into the relationship between seminoma and primordial germ cells (PGCs). By exploring this connection, we gained valuable insights into the developmental origins of seminoma, which could contribute to a deeper understanding of its pathogenesis. Lastly, our research has offered compelling evidence supporting the potential functions of TFAP2C in regulating seminoma. This finding opens up new avenues for investigating the molecular mechanisms underlying seminoma development and may hold implications for future therapeutic strategies.

In summary, our work not only reaffirms the existing knowledge about seminoma but also expands our understanding of this unique cancer type. The novel markers we identified, insights into the relationship with PGCs, and the role of TFAP2C provide valuable

contributions to the field, advancing our knowledge and potential approaches for the diagnosis and treatment of seminoma.

- Quite a number of techniques are used (like ATAC-seq to validate scRNAseq) although the additional value of those are not clear, and again the well known targets are identified;

Thank you for the feedback. We appreciate the reviewer's acknowledgment of the known targets we identified, which indeed validates the credibility of our research. However, we would like to highlight that our approach went beyond the traditional scope, employing a genome-wide unbiased analysis that allowed us to uncover new targets whose functions in seminoma were previously unknown (Fig 2E & S2D). Moreover, our study utilized single-cell ATAC-seq analysis, which provided a novel perspective on the chromatin landscape of seminoma. Through this innovative approach, we were able to unravel the pivotal role of TFAP2C in orchestrating the chromatin dynamics within seminoma cells. This specific aspect had not been thoroughly investigated before, and our findings shed new light on the potential underlying mechanisms of seminoma development.

In summary, our research not only identified known gene targets but also brought to light new targets that had not been previously associated with seminoma. Furthermore, our investigation into the role of TFAP2C through single-cell ATAC-seq analysis provided valuable mechanistic insights that were previously unexplored. We believe that our work contributes to a deeper understanding of the molecular landscape of seminoma and may pave the way for novel therapeutic strategies and potential advancements in the field.

- The study is becoming less straightforward from Figure 3 onwards. The CUT & TAG results indicate that 7210 genes are bound and regulated by TFAP2C (that is a lot);

Thank you for this comment. We appreciate your feedback, and we understand the importance of ensuring the rigor and accuracy of our analysis. The initial identification of 7210 genes bound by TFAP2C was based on relatively less stringent cutoffs (p-adjust < 0.01 & +/- 3kb of TSS as promoter regions). We opted for these cutoffs to be more inclusive and to align with previous studies that have utilized similar criteria^{1,2}. However, we acknowledge the need to maintain the robustness of our findings, and to address this, we took the initiative to re-do the CUT&Tag profiling with improved controls (IgG) and adopted more strict cutoffs (p-adjust < 10⁻⁵ & +/- 1kb of TSS as promoter regions). As a result of the updated analysis, we have now identified 4906 genes that are bound by TFAP2C, providing a more refined and rigorous gene target list. We have made the necessary revisions to the manuscript and figures to accurately reflect these changes and ensure the validity of our findings (please refer to Fig 3J).

- The representative nature of the TCam-2 cell line for seminoma is the best so far, but not absolute as reported before. In that context, performing one cell line for the KO approaches is quite thin in making a firm conclusion. Therefore the statement "Altogether, these results revealed the key role of TFAP2C in promoting migration and invasion in seminoma by directly regulating genes such as FOSL1, CCN3 and LYPD3" is premature. In addition, the

state of development to be investigated in this set up will be the transition between GCNIS and seminoma;

Thank you for your comment. Firstly, we acknowledge that our initial conclusion was primarily based on *in vitro* results, and we recognize the need to clarify this aspect in our revised manuscript. While we understand that using the TCam-2 cell line to study seminoma has its limitations in faithfully representing the tumor *in vivo*, currently, there is no better alternative available for studying seminoma. We have taken this into consideration, and in our revised manuscript, we have made the necessary changes to better articulate our findings. Specifically, we state: "Altogether, these results revealed the key role of TFAP2C in promoting migration and invasion by directly regulating genes such as *FOSL1*, *LYPD3*, and *ITGA6* in the seminoma cell line. These findings prompt further investigation to better understand the role of TFAP2C in seminoma tumorigenesis." (Please refer to Page 7, Line 36 – Line 39).

Secondly, we share the reviewer's interest in investigating the transition between GCNIS and seminoma, which is crucial for understanding the origin of GCNIS and its progression to seminoma. However, one significant obstacle we encountered was the scarcity of human GCNIS samples, which hindered the completion of this specific research. Instead, in our current work, we focused on studying human seminoma samples and conducted a comparison between PGCs and seminoma. Through this approach, we aimed to elucidate the connections between PGCs and seminoma, shedding light on the molecular features of GCNIS as the transitional cells between PGCs and seminoma.

- Related to the immune profiles, the numbers are quite limited. In addition Figure 4 is irrelevant based on the clinical behavior of seminoma (see introduction);

Thank you for your feedback. We acknowledge that our initial analysis of immune cells was not comprehensive enough. During the revision process, we have conducted additional analyses to better understand the immune cell population. This includes a combined analysis of single-cell RNA sequencing (scRNA-seq) and single-cell ATAC-seq (scATAC-seq) data to study the key transcription factors (TFs) that regulate different immune cell types (refer to Fig S4B). Additionally, we performed a detailed analysis specifically focusing on B cells (refer to Fig S4D-E).

Regarding the clinical behavior of seminoma, we observed high expression levels of *FOXP3* (a marker of regulatory T cells) and *IL21* (a gene expressed in exhausted T cells), both of which were associated with poor prognosis. We understand that the p-values obtained from our analysis may not be highly significant. We believe this is likely due to the nature of seminoma as a highly curable cancer. Nevertheless, our findings provide new insights into the potential implications of these immune markers in seminoma prognosis.

- Figure 5 and 6 as well as the related text is an overinterpretation of the data. This needs to be adjusted to be realistic related to the results shown. In addition, more validation will be needed.

Thanks to the reviewer for the comment. We have revised our interpretation to reflect the results in a more proper and realistic manner. Please see Page 9 Line 31 – Line 33 and Page 10 Line 11 – Line 19 in the revised manuscript.

We also conducted extensive validation experiments to further support our genomics findings, including staining of MMP9, CTSK and CD68, as well as MIF and CD74. Please see Fig 5G and Fig 6E in the revised manuscript.

Overall, the manuscript includes a number of interesting aspects, but besides the confirmatory data (which are well described), the more novel results are still preliminary.

Reviewer #2, expertise in epigenomics (CUT&Tag) (Remarks to the Author):

Summary:

Lu et al. performed single cell RNA-seq and ATAC-seq to characterize gene regulatory programs in various cell types from human testicular germ cell tumor samples. From functional perturbation assay, their data revealed the roles of TFAP2C in seminoma contributing to the growth, migration, and invasion programs. Furthermore, the authors identified the distinct distribution of immune cells in the seminoma microenvironment through spatial transcriptome profiling. Their results indicated that exhausted immune cells are enriched within tumor regions while effector immune cells are mainly located in the stroma. From further analysis, the authors hypothesized that tumor-derived MIF might repress immune cell functions in the seminoma microenvironment. Overall, the authors compile a multi-omics atlas dataset for testicular germ cell tumors, which would advance molecular understanding during seminoma tumorigenesis.

Major comments:

1. The major goal of this study is to characterize gene regulatory programs in various cell types from testicular germ cell tumor samples. After the authors integrated scRNA-seq data from four individuals with the scATAC-seq data generated from other human specimens, they only obtained 8 refined cell clusters containing both scRNA-seq and scATAC-seq data. However, immune cells are not included in those 8 refined cell clusters. As immune cells are the major population (>50%) within tumor tissue and play important roles during tumorigenesis, generating high quality multiome data for this diverse major population will be essential.

Thank you for your comment. We acknowledge that the immune cells represent a significant proportion of all the profiled cells, and we recognize the initial inadequacy in our analysis of immune cells. In response to this feedback, we have made improvements in our revised manuscript to better understand the roles of immune cells in seminoma. One of the major enhancements we implemented was conducting additional analysis, which involved a combined study of single-cell RNA sequencing (scRNA-seq) and single-cell ATAC-seq (scATAC-seq) data to explore the key transcription factors (TFs) involved in regulating different immune cell types (please refer to Fig R2.1). We observed consistent enrichment of binding motifs for TFs such as RUNX and EOMES in T cells^{3,4}, RUNX2, EOMES, and ETS1 in NK cells^{5,6}, PRDM1 and IRF4 in B cells, PAX5 in naïve and memory B cells, and BCL11A and IRF shared in B cells and macrophages^{3,7}. These findings align with previous work and are now included in Fig S4B in the revised manuscript. Furthermore, we also conducted new analyses to gain a deeper understanding of the roles of B cells, and these findings are now presented in Fig S4D-E. Overall, the additional analyses and experiments we have incorporated into the revised manuscript have significantly contributed to our improved comprehension of the critical roles played by immune cells in seminoma (please refer to Page 8, Line 12 – Line 15 and Line 31 – Line 39).

Figure R2.1. Enrichment of TF motifs for immune cell subtypes

2. To help assess the data quality, several key experiments/validations are necessary:

- Tn5 transposase-based assay is prone to open chromatin bias. The authors need to provide more evidence to support the specificity of CUT&Tag data generated in this manuscript.

Thank you for this comment. We appreciate your feedback, and we understand the importance of ensuring the rigor and accuracy of our analysis. The initial identification of 7210 genes bound by TFAP2C was based on relatively less stringent cutoffs (p-adjust < 0.01 & +/- 3kb of TSS as promoter regions). We opted for these cutoffs to be more inclusive and to align with previous studies that have utilized similar criteria^{1,2}. However, we acknowledge the need to maintain the robustness of our findings, and to address this, we took the initiative to re-do the CUT&Tag profiling with improved controls (IgG) and adopted more strict cutoffs (p-adjust < 10⁻⁵ & +/- 1kb of TSS as promoter regions). As a result of the updated analysis, we have now identified 4906 genes that are bound by TFAP2C,

providing a more refined and rigorous gene target list. We have made the necessary revisions to the method section and Fig 3J of the revised manuscript to accurately reflect these changes and ensure the validity of our findings.

b) Utilize orthogonal methods (immunohistochemistry or immunofluorescence staining) to verify sequencing-based results.

Thank you for your feedback. We appreciate the comment and the opportunity to provide further information regarding our research.

Through our single-cell RNA-seq analysis, we observed substantial immune cell infiltrations in the seminoma tumor microenvironment (TME). To validate this finding at the protein level, we conducted immunohistochemistry (IHC) staining for specific markers in the tumor samples, as shown in Fig 1D and IF staining in Fig S1B.

During the revision process, we expanded our analysis to include additional experiments. Firstly, we performed IHC staining for TFAP2C in four new seminoma samples, and the results are now presented in Fig 3E (Figure R2.2A). We have also made modifications to this section of the result description (please refer to Page 7 Line 2-Line 5). Furthermore, based on our analysis of T cells, we developed a hypothesis suggesting that the T cell population within the tumor exhibits a more exhausted phenotype. To support this hypothesis, we performed additional staining for PDCD1, an exhaustion marker, in four additional samples, as depicted in Figure R2.2B. Our findings confirmed a higher enrichment of exhausted cells within the tumor region, supporting our initial observations. Additionally, we re-stained MMP9, CTSK, and CD68 using the new samples, and the results showed more pronounced co-localization, as shown in Figure R2.2C. Similar results were also observed for MIF and CD74 (Figure R2.2C). These results are now included in Fig 4F, Fig 5D and Fig 6E in the revised manuscript.

Taken together, we believe that these additional experiments and analyses have further reinforced our findings and provided valuable insights into the immune cell infiltrations and exhausted T cell phenotype within the seminoma TME.

Figure R2.2. Immunostaining in seminoma

(A) IHC staining for TFAP2C in 4 seminoma samples.

(B) IHC staining for PDCD1 and CD3.

- (C) IF staining for MMP9, CTSK and macrophage marker CD68.
- (D) IF staining for MIF and CD74.

c) Verify functional interaction between tumor-derived MIF signaling and immune cells.

Thank you for your suggestion. To investigate the potential functional interaction between MIF signaling and immune cells, we conducted an *in vitro* experiment. Cultured B cells were treated with MIF for 48 hours, and we then used flow cytometry to analyze the effects of MIF stimulation on B cell activation (please refer to Page 21 Line 4-Line 19 for the detailed experimental procedure in the revised manuscript). Our results revealed a significant increase in the proportion of activated B cells following MIF treatment compared to the control (Figure R2.3). We have included these findings in the revised manuscript and figures to further support our observations and demonstrate the impact of MIF signaling on B cell activation (Please refer to Fig 6F and Page 10 Line 11-Line 19 in the revised manuscript).

Figure R2.3. MIF induces B cell activation *in vitro*

3. The lack of quantitative comparisons and statistical methods in several figures:

a) What kind of correlation analysis is used in Fig.2D, S2C?

Thank you for pointing this out. Our correlation analysis was conducted using Pearson's correlation coefficient, and we have added this information in the revised manuscript. Please see figure legend for Fig 2D and S2C, Page 13 Line 21 - Line 22 and Page 15 Line 25 - Line 26.

b) What is the scale for the heatmap shown in Fig.2E-G?

Thanks for pointing this out. We used Z-scale in the heatmap. We have added this information in the revised manuscript.

c) The expression of those key marker genes is only present in the subtype immune cells. Macrophage is also a small fraction within the tumor. Therefore, it is not appropriate to use

bulk RNA-seq data and correlate it with survival data as the p-value needs to be more significant in Fig. 4H, 5F.

Thank you for raising this interesting point. Here, we performed a comprehensive analysis to annotate subtypes of immune cells in the tumor microenvironment (TME), which is a standard approach in the field⁸⁻¹⁰. This detailed analysis allowed us to gain a better understanding of the specific roles that each immune cell subtype plays in the TME. It is widely accepted that different subtypes of immune cells, even if they are rare, can have critical functions within the TME and may be highly associated with prognosis¹¹⁻¹⁶. Regarding the discrepancy between the proportions of macrophages observed in our single-cell RNA-seq analysis and scATAC-seq analysis and IF staining results, we acknowledge that this may be due to the large size or irregular shape of macrophages, which can potentially make them less likely to be captured in single-cell sequencing techniques. However, our IF staining results supported the presence of extensive macrophage infiltration in seminoma, indicating their importance in the TME.

As for the survival analysis, we agree that the p-value obtained may not be highly significant. We believe that this is likely due to the nature of seminoma as a highly curable cancer, which has also been observed in other studies involving testicular germ cell tumors (TGCTs)^{17,18}. The generally favorable prognosis of seminoma could lead to less pronounced statistical significance in survival analysis.

d) No quantification results in Fig. 3J, 5G, 6E, S3G.

Thanks for your suggestion. We quantified these results and showed in Fig 3J (please refer to Figure R2.4A below), 5G (please refer to Figure R2.4C below), 6E (please refer to Figure R2.4D below), and S3G (please refer to Figure R2.4B below) in the revised manuscript.

Figure R2.4. Quantification results

- (A) The quantification for Fig 3J
 (B) The quantification for Fig S3G
 (C) The quantification for Fig 5G
 (D) The quantification for Fig 6E

Minor comments:

1. Please provide a scale bar and low-magnification images to illustrate the distribution of those markers within tumor sections in Fig. 1D.

Thanks to your suggestion. We re-stained the marker genes in Fig 1D using new patient samples and took 20X images showed in the Fig 1D in the revised manuscript.

2. Mislabeled GO terms in Fig. 3H; the red box should represent “Upregulated GO terms in siTFAP2C”.

Thank you for pointing out this mistake. We have revised this part accordingly in Fig 3H.

3. Include N in survival analysis Fig.4H.

Thank you for your comment. The sample number (n=133) is presented in the bottom panel of Fig 4H.

4. MMP9 and CTSK don't seem colocalized in Fig.5D.

Thank you for your comment. We acknowledged that our initial staining for MMP9 and CTSK were not ideal. During the revision, we conducted additional staining for MMP9 and CTSK with new samples, through which we can observe better co-localization of them with CD68. The new staining results are now included in Fig 5D.

Figure R2.5. IF staining for MMP9, CTSK and CD68. The white arrows represented colocalization between MMP9 and CD68, or CTSK and CD68, respectively.

5. Please provide a scale bar in Fig. 5G, 6E.

Thank you for your comment. We have revised Fig 5G and 6E to include the scale bars.

6. Please provide more information on how CNV information is generated from single cell RNA-seq data in Fig. S1D?

Thank you for pointing this out. We have included the information in the method section of the revised manuscript. Please see Page 25 Line 27-Line 33.

7. Lack of group information in Fig. S3H.

Thank you for pointing this out. We have included the information in Fig S3H in the revised manuscript.

8. Data presentation should be consistent. For instance, tissue sample 1 & 3 were used to illustrate the location of immune cells in Fig. 4F. But tissue sample 2 was used in Fig. 5D. In addition, two samples were shown in Fig. S4E but three samples were shown in S4F.

Thank you for your comment. Since we conducted spatial transcriptomic profiling with

three biological replicates, we encountered limitations in accommodating all the results within the main figure due to space constraints. Consequently, we decided to incorporate an overall quantification of the co-localization between MMP9, CTSK, and macrophages in the main figure (Fig 5D), while placing the detailed spatial expression patterns in the supplementary figure (Fig S5A). This approach allows us to present the essential findings in the main figure while ensuring that the complete spatial expression data is available for reference in the supplementary figure. We have also made modifications in the result description accordingly (please see Page 9 Line 23-Line 25).

Fig R2.6. The expression correlation of *MMP9*, *CTSK* and Macrophage in spatial transcriptome data. *CD68* and *ACTB* as positive control and negative control, respectively.

Reviewer #3, expertise in single cell multi-omics and spatial transcriptomics (Remarks to the Author):

In their research, Lu and colleagues conducted a comprehensive exploration of molecular characteristics and microenvironmental aspects using single-cell RNA-seq and ATAC-seq and spatial transcriptomics within seminoma patient specimens and cell lines. Through genomic profiling and functional comparative studies between seminoma cells and primordial germ cells, the team discovered critical gene expression programs, prominently featuring the gene TFAP2C, as well as 15 distinct immune cell subtypes in the tumor microenvironment (TME), each with unique spatial localizations. Overall, this study significantly extends our understanding of seminoma pathogenesis and illuminates potential therapeutic targets by establishing a multi-omics atlas. While the data fundamentally corroborates the propositions presented, there are several aspects that could benefit from further refinement and improvement. More specifically:

Major

1. The authors claimed that they provided the atlas for the seminoma, yet their study only encompasses four samples. This limited sample size may hinder the universality of the findings. To enhance data reliability, the reviewer advises the researchers to expand their sample size.

Thank you for raising this important point. We fully recognize the significance of having an adequate number of patients to establish robust and reliable conclusions in our study. Here, we want to emphasize that we utilized samples from more than 4 patients in our research. Specifically, we employed samples from 4 patients for single-cell RNA-seq profiling, 3 patients for spatial transcriptomics profiling, 1 patient for single-cell ATAC-seq profiling, and 6 patients for IHC/IF staining confirmation. In total, we analyzed samples from 14 different individuals in this study, and consistently observed patterns among them. By utilizing a sufficient number of patients and multiple approaches, we aimed to ensure the reliability and reproducibility of our findings. We are confident that our study design and the number of patient samples examined provide strong support for the conclusions drawn.

2. The “Immune Cell Infiltration in Seminoma” section aims to further break down the immune cell clusters in Figure 1B. This section heavily focuses on the extensive analysis of T cells, providing little breakdown of the B cells and their role in the seminoma microenvironment. This section could be further improved by incorporating more analysis on the functionality of B cell subtypes (marginal zone, memory, naïve B cells) in the TME.

Thank you for the suggestion. We acknowledge the importance of gaining a deeper understanding of the roles of B cells in the tumor microenvironment (TME). In response to your feedback, we performed a focused analysis of B cells during the revision process, which allowed us to identify three distinct B cell subtypes: naïve B cells, memory B cells, and plasma cells. Through pseudotime analysis, we observed a developmental trajectory

across these subtypes, shedding light on their potential functions and relationships (Figure R3.1A). Furthermore, downstream analysis helped us identify enriched Gene Ontology (GO) terms in each B cell subtype, such as "antigen processing and presentation," "immune response," and "B cell activation" (Figure R3.1B). To delve deeper into the regulatory mechanisms underlying B cell subtypes, we conducted a combined analysis of single-cell RNA-seq and single-cell ATAC-seq data, enabling us to identify key transcription factor (TF) motifs enriched in each B cell subtype (Figure R3.1C). Notably, these findings were highly consistent with previous work (Figure R3.1D-E)³, further validating the reliability and relevance of our results. The new analysis has been incorporated into Fig S4D-E in the revised manuscript (please see Page 8 Line 31-Line 39), bolstering the comprehensiveness and significance of our study in understanding the functional diversity of B cells within the tumor microenvironment.

Fig R3.1. B cell clusters in seminoma

- (A) Integrative lineage trajectory of B cell subtypes in seminoma. The smoothed line and arrow represent the visualization of the trajectory path across different subtypes (naive, memory and plasma).
- (B) GO terms enriched in B cell subtypes.
- (C) TF binding motifs enriched in immune cell subtypes.
- (D, E) B cell pseudotime analysis of the data in You's study ³.

3. In Figure 2, the correlations between PGCs and seminoma are discussed, however there is some positive correlation. There is some similarity between the fetal SSCs state 0 and seminoma in the PCA analysis (2C) and the heatmap in 2D that is not discussed. This relationship should also be considered and discussed. Additionally, the Figure 2F Adult SSC states are not very informative as it is depicted. Since the authors are focusing on the distinctions between seminoma and PGC's, the adult SSCs could be removed from this figure.

Thank you for your valuable suggestion. In our previous work^{19,20}, we presented a model describing male germline development from the fetal stage to adult spermatogonia, which involves the following stages: PGC -> State f0 -> State 0 -> State 1 -> State 2-4. As seminoma is a type of germ cell tumor found in the adult human testis, we performed a comparison with all adult spermatogonia states, as well as PGCs and State f0. Both principal component analysis (PCA) and correlation analysis indicated the highest similarities between PGCs and seminoma (R=0.89) (Figure R3.2A). As the reviewer rightly pointed out, seminoma also exhibited some similarity with State f0 (R=0.74) (Figure R3.2A). However, this observation is not surprising, as State f0 directly differentiates from PGCs (R=0.79) (Figure R3.2A). Considering that many key markers for seminoma (such as TFAP2C, POU5F1, and SOX17) are expressed in PGCs but selectively repressed in State f0 (Figure R3.2B), we believe it is premature to establish strong connections between State f0 and seminoma.

Regarding the suggestion to remove adult SSCs from this Figure, we appreciate the input as it would help streamline the analysis. However, as mentioned above, our intention was to compare seminoma with human male germ cells at various developmental stages and determine which stage(s) it closely resembles. After conducting the analysis, we reached the conclusion that seminoma is highly similar to PGCs. Thus, we believe that including the adult spermatogonia analysis is beneficial and informative for understanding the developmental relationships of seminoma.

Fig R3.2. Correlation among seminoma tumor cells and germ cells in early stage
 (A) Correlation analysis among seminoma tumor cells and germ cells in early stage using Pearson's correlation coefficients.
 (B) Expression of *TFAP2C*, *POU5F1*, *NANOG* and *SOX17* in seminoma tumor cells and

germ cells in early stages. The white dots represent mean expression.

4. In Figure 2E, the authors point out the shared specific expression of genes between seminoma and PGCs (TFAP2C, POU5F1, NANOG and SOX17). Figure 2E also has a very interesting cluster 3 with some genes that are upregulated in PGCs yet downregulated in seminoma. Similarly, cluster 1 has some genes that are enriched in seminoma but not PGCs. Since this section focuses on the analysis comparison between seminoma and PGCs, this would be important to discuss and enhance the analysis of this data.

Thank you for your suggestion. We performed additional Gene Ontology (GO) functional annotation analysis on the genes specific to seminoma and PGCs, specifically those in Cluster 1 and Cluster 3 in Fig 2E. We found that seminoma-specific genes were mainly enriched in pathways such as "Phagocytosis", "B cell receptor signaling pathway", and "Innate immune response". In contrast, PGC-specific genes were primarily enriched in pathways involved in "male gonad development", "response to retinoic acid", and "stem cell fate specification" (Figure R3.2A). The new analysis is now included in Fig S2D and Page 6 Line 10-Line 15 in the revised manuscript. Please note that these findings are consistent with our previous results (Figure R3.2B) ²⁰.

Fig R3.3. Specific genes expression and GO terms enriched in seminoma and PGCs.

(A) Specific genes and GO terms enriched in seminoma and PGCs.

(B) GO terms enriched in PGCs in Guo's data²⁰.

5. Figure S2A and Figure 2E, it is not clear the numbering system of the clusters. Figure 2E has 5 clusters, but there are more than 5 clusters shown in Figure S2A.

Thank you for your comment. We want to clarify that the cell types in Figure 2E correspond to the cell types in Figure S2A, presented in the same order and color, which are shown on the top of the heatmap.

Regarding the GO terms shown on the right side of the heatmap, they represent gene

modules that are expressed in the eight cell types and are enriched for specific biological functions. We understand that this may have been misinterpreted due to our color settings. To rectify this, we have made the necessary adjustments in Fig 2E in the revised manuscript to ensure clarity and accuracy in presenting the information.

6. In the text, the authors describe the use of spatial transcriptomics profiling for this dataset – suggesting that *TFAP2C* and *SOX17* are highly expressed within areas of seminoma in the spatial data. It is not clear how the author defines the areas of seminoma as Figure 2G and S2 G using just *TFAP2C* and *SOX17*, when before this, the author mentions marker genes for seminoma as *POU5F1+*, *TFAP2C+* and *NANOG+*. To keep things consistent, it would be useful to have the spatial regions containing seminoma defined by these markers.

Thank you for your comment. We would like to clarify that the identification of seminoma regions in our study was not solely based on the expression of a few selected genes. To define cell types in the spatial transcriptomics data, we employed a robust approach by integrating scRNA-seq data with the spatial transcriptomics data. By transferring the cell labels from the scRNA-seq analysis to annotate the spatial transcriptomics data, we followed the standard procedure for cell type annotation in spatial transcriptomics studies. In addition, we chose to display the expression patterns of key marker genes, such as *TFAP2C*, *SOX17*, and *NANOG* (Figure R3.4A). These genes are widely recognized as highly representative markers for seminoma and play a crucial role in defining the identity of the tumor. The results of this integration can be found in Fig 4E and Fig S4F in the revised manuscript. Through this combined approach, which involves integrating annotations from scRNA-seq and the expression of marker genes, we were able to collectively identify the spatial localization of tumor cells within the spatial groups (Figure R3.4B).

Fig R3.4. Tumor marker gene expression and scRNA-seq annotation in spatial transcriptome data.

- (A) Seminoma tumor marker (*TFAP2C*, *SOX17* and *NANOG*) expression location in spatial transcriptome data.
- (B) Location of seminoma tumor cell annotation with scRNA-seq data in spatial transcriptome data.

7. The healthy tissue and tumor tissue have very distinct clusters in the Figure 3A, does it mean there is very little cell type overlap between normal tissue and the seminoma tissue? Further, there is no labeling of the colors depicted in these clusters, which are necessary for the reader to interpret this data. According to the markers and corresponding legend in 3C: *TFAP2C*, *POU5F1* and *SOX17* are markers of seminoma, yet this seems to only apply to the top left cluster. Can the author clarify why the tumor clusters in 3A are inclusive of more than just that top left cluster? The authors should also annotate the clusters in Figure 3B, the way it is at present is uninformative.

Thank you for your suggestion. We apologize for any confusion caused. We have taken your feedback into consideration and made the following improvements. Firstly, we have revised the cell type annotation for the scATAC-seq data to make it more clear and understandable (Figure R3.5A). Regarding the scRNA-seq data, we conducted a detailed analysis and observed significant differences in cell types between seminoma tissue and normal tissue (Figure R3.5A and R3.5B). In the tumor samples, there is a predominance of lymphocytes and tumor cells (Figure R3.5C), while healthy testicular tissue mainly consists of germ cells and somatic cells, with minimal presence of immune cells (Figure R3.5D)²¹. Due to these substantial differences, there is very little overlap in cell types between the two groups. Please refer to Fig 3A-B and Fig 1B-C.

Fig R3.5. Cell types in seminoma and healthy testis.

- (A) Cell types in scATAC-seq data of seminoma patient sample and health sample.
- (B) Samples distribution of scATAC-seq data show in A.
- (C) Cell types in seminoma scRNA-seq data.
- (D) Cell types in healthy adult whole testes scRNA-seq data from Guo's data²¹.

8. In addition, Figure 5F appears to be somewhat ambiguous and perplexing in its current presentation. It is unclear from the visual data representation which curve corresponds to gene expression and which one signifies the survival rate. To improve readability and comprehension, it would be beneficial to delineate these two components more explicitly, perhaps through distinct labeling, color coding, or other methods of differentiation. This would ensure the accurate interpretation of the figure and its corresponding data.

Thank you for pointing this out. We have added the labeling information in the revised manuscript. Please see Fig 4G and Fig 5F.

Minor

1. In Figure 6E, are the IF images adjacent to those in 6D? If so, please depict the ROI in which the IF was done. If not, this data would be more cohesive if the IF staining was done on a section adjacent to the tissues that the spatial transcriptomics analysis was done on.

We greatly appreciate this viewpoint. However, we regretted that due to the slicing process after sample collection, we were unable to preserve these adjacent sections intact. In future studies, we plan to follow this approach in order to further investigate the unresolved issues mentioned in this article.

2. Figure S2F contains informative knowledge of the spatial clustering of the tumor tissue sections. The paragraph corresponding to this section mentions the gene expression profiles of each cluster. Since this is spatial data where visualization of clusters is important, I would suggest visualizing this data in Figure 2. It would also be useful to have annotations of the clusters on Figure S2F.

Thank you for your suggestion. We utilized gene expression and scRNA-seq annotation to determine the approximate location of the distribution of each cell type in the spatial group data, and this result is shown in Fig S2F of the revised manuscript.

3. The authors describe cytotoxicity-related genes twice on page 7 (line 46) and page 8 (line 29). It would be useful to explain the selection of these genes in both instances.

Thanks for your suggestion. We added the description of this section in the revised manuscript. Please see Page 8 Line 45 – Page 9 Line 3.

4. Figure 3J depicts the multiple modalities that are highlighted in this study and thus should

have a more extensive analysis.

Thanks for your suggestion. We identified the gene expression programs that were up- or down-regulated by TFAP2C through the TFAP2C knockdown experiment (See Figure 3I). We also discovered genes that were bound by TFAP2C through the CUT&Tag experiment. Combined analysis helped identify genes that were directly suppressed by TFAP2C (n=346) through intersecting downregulated genes in si*TFAP2C* group (n=843) with TFAP2C bound genes (n=4906), and the associated GO terms (See Figure 3J). The original Figure 3J (now Figure 3K in the revised manuscript) showed the coverage plots of the representative genes which are directly downregulated by TFAP2C, which is a direct visual display of the genomics results. We have articulated this more clearly in our revised manuscript.

5. There are also some minor issues regarding the content that could be modified. For instance, in the CUT&Tag library generation section under the methods category, the cell numbers were incorrectly recorded as 1x10⁵, rather than the correct form, 1x10⁵.

Thank you for pointing out this error. We have corrected it in the revised manuscript. Please see Page 22 Line 2.

REFERENCES

1. Liu, Z. *et al.* RNA Helicase DHX37 Facilitates Liver Cancer Progression by Cooperating with PLRG1 to Drive Superenhancer-Mediated Transcription of Cyclin D1. *Cancer Res* **82**, 1937-1952 (2022).
2. Yuan, S. *et al.* The histone modification reader ZCWPW1 promotes double-strand break repair by regulating cross-talk of histone modifications and chromatin accessibility at meiotic hotspots. *Genome Biol* **23**, 187 (2022).
3. You, M. *et al.* Single-cell epigenomic landscape of peripheral immune cells reveals establishment of trained immunity in individuals convalescing from COVID-19. *Nat Cell Biol* **23**, 620-630 (2021).
4. Zheng, L. *et al.* Pan-cancer single-cell landscape of tumor-infiltrating T cells. *Science* **374**, abe6474 (2021).
5. Li, K. *et al.* Landscape and Dynamics of the Transcriptional Regulatory Network During Natural Killer Cell Differentiation. *Genomics Proteomics Bioinformatics* **18**, 501-515 (2020).
6. Wang, D. & Malarkannan, S. Transcriptional Regulation of Natural Killer Cell Development and Functions. *Cancers (Basel)* **12**(2020).
7. Satpathy, A.T. *et al.* Massively parallel single-cell chromatin landscapes of human immune cell development and intratumoral T cell exhaustion. *Nat Biotechnol* **37**, 925-936 (2019).
8. Gong, L. *et al.* Comprehensive single-cell sequencing reveals the stromal dynamics and tumor-specific characteristics in the microenvironment of nasopharyngeal carcinoma. *Nat Commun* **12**, 1540 (2021).
9. Guo, X. *et al.* Global characterization of T cells in non-small-cell lung cancer by single-cell sequencing. *Nat Med* **24**, 978-985 (2018).
10. Sun, Y. *et al.* Single-cell landscape of the ecosystem in early-relapse hepatocellular carcinoma. *Cell* **184**, 404-421 e16 (2021).
11. Bonnal, R.J.P. *et al.* Clonally expanded EOMES(+) Tr1-like cells in primary and metastatic tumors are associated with disease progression. *Nat Immunol* **22**, 735-745 (2021).
12. Cheng, S. *et al.* A pan-cancer single-cell transcriptional atlas of tumor infiltrating myeloid cells. *Cell* **184**, 792-809 e23 (2021).
13. Hu, S. *et al.* TDO2+ myofibroblasts mediate immune suppression in malignant transformation of squamous cell carcinoma. *J Clin Invest* **132**(2022).
14. Jiang, M. *et al.* FOXP3-based immune risk model for recurrence prediction in small-cell lung cancer at stages I-III. *J Immunother Cancer* **9**(2021).
15. Qi, J. *et al.* Single-cell and spatial analysis reveal interaction of FAP(+) fibroblasts and SPP1(+) macrophages in colorectal cancer. *Nat Commun* **13**, 1742 (2022).
16. Yang, S. *et al.* FOXP3 promotes tumor growth and metastasis by activating Wnt/beta-catenin signaling pathway and EMT in non-small cell lung cancer. *Mol Cancer* **16**, 124 (2017).
17. Ji, C. *et al.* Immune-related genes play an important role in the prognosis of patients with testicular germ cell tumor. *Ann Transl Med* **8**, 866 (2020).

18. Takami, H. *et al.* Transcriptome and methylome analysis of CNS germ cell tumor finds its cell-of-origin in embryogenesis and reveals shared similarities with testicular counterparts. *Neuro Oncol* **24**, 1246-1258 (2022).
19. Guo, J. *et al.* Chromatin and Single-Cell RNA-Seq Profiling Reveal Dynamic Signaling and Metabolic Transitions during Human Spermatogonial Stem Cell Development. *Cell Stem Cell* **21**, 533-546 e6 (2017).
20. Guo, J. *et al.* Single-cell analysis of the developing human testis reveals somatic niche cell specification and fetal germline stem cell establishment. *Cell Stem Cell* **28**, 764-778 e4 (2021).
21. Guo, J. *et al.* The adult human testis transcriptional cell atlas. *Cell Res* **28**, 1141-1157 (2018).

Reviewers' Comments:

Reviewer #2:

Remarks to the Author:

The authors have addressed the comments and provided additional functional validations during this revision. This research is now suitable to be considered for publication.

Reviewer #4:

Remarks to the Author:

In the revised manuscript, the authors have adequately addressed review#3's concerns.

There are a few additional places that requires minor clarification and correction.

1. In Fig. 3I and J, there are three places with "p-value" written, e.g., Fig.3J, "GO terms enriched (p-value)". It is unclear what does this mean: does the authors intend to denote p-value cut off, i.e, p-value < 0.05 but missed the latter part (" <0.05 ")?
2. In Fig. 3I and J, the authors should show the actual enrichment p-values of the GO terms listed (e.g., cell adhesion).
3. On page 25, line 1, the authors used the Scanorama algorithm to integrate spatial and single cell RNA-seq data, but did not add reference to the Scanorama paper.

Reviewer #5:

Remarks to the Author:

In this study, Lu and colleagues describe single cell RNA sequencing and spatial transcriptomics of four primary seminoma samples combined with analysis of the only available seminoma cell line, TCAM-2. This study builds on the one published single cell analysis paper from another group in 2022 (which was of only one sample), suggests that TFAP2C is an expressed gene essential for the germ cell tumor phenotype by promoting genes associated with invasion relative to non-cancerous germ cells, and identifies a tumor-associated immune infiltrate that displays features of exhaustion.

Overall I find the intent of the study, to try and elucidate the molecular determinants of seminoma initiation and oncogenic behavior, to be important. I believe the authors are to be congratulated for implementing their multi-omics approach including single cell analysis and cell line validation efforts. The mechanistic studies around TFAP2C, a gene known to be expressed in seminoma from prior work, are interesting and I believe that the authors used the available in vitro models (which are very limited) to the degree possible. I also especially find the immune cell findings of interest, though their overall significance on the clinical behavior of seminoma are less clear.

Overall, I am also unclear that the findings herein provide important new markers of seminoma (the diagnosis of localized seminoma is not particularly difficult using known markers), or mechanisms into how seminoma relapses (it is not clear that any of the patients whose tissue was used for this study received chemotherapy or experienced relapsed).

The first question I was asked to address was whether the authors sufficiently responded to the original Reviewer #1 comments. From my take, reviewer #1 mostly questioned the novelty of the findings, and I tend to agree with their initial review points. Aside from a few specifics around the immune cell compartment, I wonder if bulk RNA analysis of these samples would have sufficed to identify TFAP2C and its downstream targets as potential mediators of seminoma behavior.

A few additional points:

The discussion around Figure 1 obfuscates the low number of actual seminoma cells isolated by single

cell RNA sequencing. Actual numbers of seminoma cells should be provided such that the reader can weigh the significance of the scRNA findings. As best I can tell, the bulk of the seminoma cell isolated were from the fourth sample, which is somewhat limiting for the generalizability of this study.

Despite my inclination to believe that seminoma closely resembles PGCs based on prior published literature, I find interpretation of Figure 2 to be difficult due to limited explanation of how data from the four seminoma samples were integrated computationally with the adult germ cell data.

Additionally, Figure 2C includes PCA3 and PCA1. Why is PCA2 not shown? This raises concern that PCA2 might change perception of results to show that seminoma is not so clearly similar to PGCs.

CellChat analysis identifies numerous receptor-ligands as enriched in single cell data. The rationale or significance of uniquely focusing on MIF in this manuscript is unclear. Are the macrophages that the MIF expressing seminoma cells are communicating with immunosuppressive (perhaps more M2 polarized - acknowledging that M1 vs M2 is an arbitrary dichotomy)

Were TFAP2C cut&tag peaks presumed to regulate their closest gene?

Peak on p. 24 is misspelled as peck

The methods section starting in the section "Cell culture and small interfering RNAs" through "Cell invasion" reads like a protocol rather than a description of what was actually done. This should be modified to reflect what was actually done.

Authors' Response to Reviewers for NCOMMS-23-15034-A by Lu et al.

Reviewer #2 (Remarks to the Author):

The authors have addressed the comments and provided additional functional validations during this revision. This research is now suitable to be considered for publication.

Reviewer #4 (Remarks to the Author):

In the revised manuscript, the authors have adequately addressed reviewer #3's concerns.

There are a few additional places that requires minor clarification and correction.

1. In Fig. 3I and J, there are three places with "p-value" written, e.g., Fig.3J, "GO terms enriched (p-value)". It is unclear what does this mean: does the authors intend to denote p-value cut off, i.e, p-value < 0.05 but missed the latter part (" <0.05 ")?

Thank you for pointing this out. As you correctly noted, our intention was to emphasize the significance of the pathway indicated, with a p-value below 0.05. We agree with the reviewer's assessment that the initial presentation was confusing. To address this, we have revised the figures accordingly. Please refer to Figure R4.1 for the updated panels, which are now presented in Figures 3I and 3J within the revised manuscript.

Figure R4.1 GO terms enriched in TCam-2 cells in vitro *TFAP2C* KD experiment

- (A) GO terms enriched in downregulated genes in si*TFAP2C* group compared to control. The length of the bar represents the value of $-\log_{10}(\text{p-value})$.
- (B) GO terms enriched in upregulated genes in si*TFAP2C* group compared to control. The length of the bar represents the value of $-\log_{10}(\text{p-value})$.
- (C) Enrichment of GO terms for genes directly targeted by *TFAP2C* in TCam-2 cells. The length of the bar represents the value of $-\log_{10}(\text{p-value})$.

2. In Fig. 3I and J, the authors should show the actual enrichment p-values of the GO terms

listed (e.g., cell adhesion).

We appreciate your suggestion. We've redrawn the enriched GO terms plot along with their corresponding p-values, as depicted in Figure R4.1.

3. On page 25, line 1, the authors used the Scanorama algorithm to integrate spatial and single cell RNA-seq data, but did not add reference to the Scanorama paper.

Thanks for pointing out this mistake. We have cited the Scanorama paper in the revised manuscript. Please refer to Page 25 Line 21 in the revised manuscript.

Reviewer #5 (Remarks to the Author):

In this study, Lu and colleagues describe single cell RNA sequencing and spatial transcriptomics of four primary seminoma samples combined with analysis of the only available seminoma cell line, TCAM-2. This study builds on the one published single cell analysis paper from another group in 2022 (which was of only one sample), suggests that TFAP2C is an expressed gene essential for the germ cell tumor phenotype by promoting genes associated with invasion relative to non-cancerous germ cells, and identifies a tumor-associated immune infiltrate that displays features of exhaustion.

Overall I find the intent of the study, to try and elucidate the molecular determinants of seminoma initiation and oncogenic behavior, to be important. I believe the authors are to be congratulated for implementing their multi-omics approach including single cell analysis and cell line validation efforts. The mechanistic studies around TFAP2C, a gene known to be expressed in seminoma from prior work, are interesting and I believe that the authors used the available in vitro models (which are very limited) to the degree possible. I also especially find the immune cell findings of interest, though their overall significance on the clinical behavior of seminoma are less clear.

Overall, I am also unclear that the findings herein provide important new markers of seminoma (the diagnosis of localized seminoma is not particularly difficult using known markers), or mechanisms into how seminoma relapses (it is not clear that any of the patients whose tissue was used for this study received chemotherapy or experienced relapsed).

We would like to express our gratitude to this reviewer for their overall appreciation of our work and the invaluable insights that have significantly guided the revision of our manuscript. First, regarding new markers for seminoma, in our research, we employed a multi-omics approach to comprehensively investigate the molecular characteristics of seminoma in a systematic fashion. This approach allowed us to confirm numerous well-established aspects of seminoma, such as *TFAP2C* and *POU5F1/OCT4*, as well as unveil several novel markers specific to seminoma, including *DPPA5*, *KHDC3L*, and *PRAME* (as illustrated in Fig 2E & S2D). These discoveries hold the potential to advance

diagnostic accuracy and precision in the future, thereby enhancing patient outcomes. Currently, we are conducting follow-up studies to further elucidate the roles of these markers in seminoma.

We also concur with the reviewer's observation regarding the high interest in understanding the mechanisms of seminoma relapse. However, as the reviewer may be aware, the current understanding of seminoma remains relatively limited, given the unique nature of this tumor type. As one of the pioneering efforts to unravel the molecular features of seminoma, our initial focus was on primary seminoma in situ, prior to chemotherapy treatment, with the aim of establishing a foundational dataset for future investigations. We have now embarked on collaborative efforts with cancer hospitals to collect samples from relapsed seminoma cases for comparative studies. These endeavors are expected to shed light on the underlying mechanisms of seminoma relapse and contribute to a more comprehensive understanding of this critical aspect of the disease.

The first question I was asked to address was whether the authors sufficiently responded to the original Reviewer #1 comments. From my take, reviewer #1 mostly questioned the novelty of the findings, and I tend to agree with their initial review points. Aside from a few specifics around the immune cell compartment, I wonder if bulk RNA analysis of these samples would have sufficed to identify TFAP2C and its downstream targets as potential mediators of seminoma behavior.

Thank you for your comment. Firstly, as you may have already observed, our analysis identified a substantial number of immune cells within the tumor samples, while the seminoma cells were relatively scarce. Conducting bulk RNA-seq in this context would result in the mixing of seminoma cell signals with strong immune cell signals, making it challenging to discern the specific seminoma cell characteristics. This issue is a common challenge in the study of various tumor types. Leveraging scRNA-seq allows us to effectively segregate the seminoma cells from the surrounding immune cells and unveil the distinctive features unique to the tumor. Hence, we believe that scRNA-seq is indispensable in this study and cannot be simply substituted with bulk RNA-seq.

Secondly, we would like to clarify that our identification of potential targets of TFAP2C was a result of a comprehensive analysis that combined scRNA-seq with other experiments, including scATAC-seq, CUT&Tag, and TFAP2C knockdown (KD) in vitro. Our approach began with the confirmation of TFAP2C-specific expression in seminoma through scRNA-seq (as shown in Figure 1F). Subsequently, the enrichment of the TFAP2C binding motif in seminoma cells was established using scATAC-seq, indicating the functional relevance of TFAP2C in seminoma (as illustrated in Figure 3D). Through CUT&Tag experiments, we analyzed the direct binding sites of TFAP2C in seminoma cell lines, as well as the genes proximal to these binding sites (as depicted in Figure 3J). Lastly, we validated the genes affected by TFAP2C through TFAP2C KD in TCam-2 cells (as shown in Figure 3F and 3I). By cross-referencing these findings with genes bound by

TFAP2C, we generated a list of candidate genes that may be directly regulated by TFAP2C (as presented in Figure 3J).

A few additional points:

The discussion around Figure 1 obfuscates the low number of actual seminoma cells isolated by single cell RNA sequencing. Actual numbers of seminoma cells should be provided such that the reader can weigh the significance of the scRNA findings. As best I can tell, the bulk of the seminoma cell isolated were from the fourth sample, which is somewhat limiting for the generalizability of this study.

We appreciate your suggestion. In our study, we successfully isolated a total of 375 tumor cells from samples obtained from four individual patients. To address your concerns about the variation in cell numbers across the four samples, we have provided a bar plot displaying the cell count for each cell type in these samples (please refer to Figure R5.1A). Additionally, we have incorporated the tumor cell count into the revised manuscript, which can be found on Page 5, Lines 32 to 34. You can also locate the aforementioned plot in Figure S1B of the revised manuscript.

Moreover, as you correctly pointed out, there are discrepancies in the number of tumor cells between the four samples, with Tumor #4 and #2 contributing a significant portion of these cells. To ensure that the molecular signatures of the tumor cells remain unaffected by these variations in cell numbers, we conducted a Principal Component Analysis (PCA). For more detailed information on this analysis, please refer to the "Principal Component Analysis (PCA)" section on Page 26, Lines 12 to 22. The results of the PCA demonstrate that the tumor cells closely cluster together, affirming the robustness and effectiveness of our scRNA-seq profiling and analysis. This analysis is now included in Figure S1C of the revised manuscript.

Figure R5.1 The cell number and similarity of cell types among samples
 (A) Bar plot showing the cell number of each cell type in four samples.

(B) PCA illustrated the similarity of gene expression levels for each cell type across four samples. Different colors represent cell types, while shapes represent different samples. The red circle marks tumor cells.

Despite my inclination to believe that seminoma closely resembles PGCs based on prior published literature, I find interpretation of Figure 2 to be difficult due to limited explanation of how data from the four seminoma samples were integrated computationally with the adult germ cell data.

Thank you for your comment. Given that seminoma is typically found in the adult testis, we decided to combine tumor cells with adult germ cells for our analysis. In doing so, we aimed to compare spermatogonia in adults with tumor cells to show that, despite their occurrence in the adult testis, the transcriptome of seminoma closely resembles that of primordial germ cells (PGCs) in the human embryo.

For our single-cell RNA sequencing analysis, we employed the Seurat package in R. This method allowed us to aggregate data from the four samples into a unified Seurat object, facilitating subsequent analyses. To distinguish tumor cells, we employed the subset function, and we similarly obtained data pertaining to adult germ cells. Following this, we integrated these datasets using the merge function and conducted re-clustering through uniform manifold approximation and projection (UMAP). Subsequently, we were able to compare gene expression patterns across these datasets. This approach is a widely accepted and commonly used method in the field of single-cell RNA analysis¹⁻³. For a comprehensive breakdown of the steps involved, please refer to the "Processing of single cell RNA-seq data" section on Page 24, from Line 3 to Line 20, in the revised manuscript.

Additionally, Figure 2C includes PCA3 and PCA1. Why is PCA2 not shown? This raises concern that PCA2 might change perception of results to show that seminoma is not so clearly similar to PGCs.

Thanks for your comment. Our analysis has unveiled similarities in gene expression between tumor cells and primordial germ cells (PGCs), as demonstrated through Principal Component Analysis (PCA) and correlation analysis. This similarity is evident in both PCA1/PCA3 and PCA1/PCA2, with PCA1/PCA3 exhibiting a higher degree of significance. Given that PCA2 and PCA3 contribute similarly to the overall patterns (16.4% vs. 10.4%), we have opted to use PCA3 for improved clarity in our presentation, as it is a widely accepted and utilized visualization approach⁴⁻⁷. To enhance the presentation of this result, we have introduced a 3D PCA plot in Figure 2C within the revised manuscript. This 3D visualization offers a more distinct representation of the relationship between seminoma and PGCs, aiding in a better understanding of this connection (as depicted in Figure R5.2).

Figure R5.2 3D plot of PCA showing the relationship between seminoma tumor cells and early germ cells.

CellChat analysis identifies numerous receptor-ligands as enriched in single cell data. The rationale or significance of uniquely focusing on MIF in this manuscript is unclear. Are the macrophages that the MIF expressing seminoma cells are communicating with immunosuppressive (perhaps more M2 polarized - acknowledging that M1 vs M2 is an arbitrary dichotomy)

Thanks for your comment. Using CellChat, we conducted an analysis to explore the interactions occurring between tumor cells and various cell types within the seminoma tumor microenvironment. Our investigation revealed that the MIF signaling pathway emerged as the most significant, with tumor cells displaying the highest expression of MIF (as depicted in Fig 6A-C). As previous research has indicated, MIF produced by tumors has been associated with inhibitory effects on the tumor microenvironment⁸⁻¹⁰. Consequently, we directed our attention toward investigating the role of MIF in the context of TGCT.

Additionally, in response to the reviewer's valuable suggestion, we carried out a comparison of the enrichment levels of M1 and M2 characteristics in macrophages. This analysis revealed a notable enrichment of M2 characteristics in macrophages, underscoring that macrophages within the TGCT microenvironment indeed exhibit an immunosuppressive state (as illustrated in Figure R5.3). We have incorporated this result into in Figure S6E the revised manuscript (please see Page 10, Lines 18-23).

Figure R5.3 Enrichment of M1 and M2 score in macrophages

Were TFAP2C cut&tag peaks presumed to regulate their closest gene?

Thanks for your comment. CUT&Tag is a chromatin profiling technique employed to examine protein-DNA interactions on a genome-wide scale. In CUT&Tag experiments, regions with modified histones and transcription factors (TFs) are recognized as collections of aligned reads, commonly referred to as "peaks" (as documented in reference 11). These peaks are often found in close proximity to the promoters of nearby genes, leading to the presumption that these genes may be potential targets of the TFs under investigation^{12,13}. However, the distance between a peak and a gene does not inherently imply a direct regulatory relationship. In some instances, regulatory elements can be situated at considerable distances from the genes they influence, or they may govern multiple genes simultaneously¹⁴. Therefore, determining the precise regulatory impact of identified peaks on neighboring genes often necessitates additional functional studies, validation experiments, and bioinformatics analyses¹⁵. In our study, we identified potential targets of TFAP2C through a comprehensive analysis that combined scRNA-seq with other experiments, including scATAC-seq, CUT&Tag, and *TFAP2C* knockdown (KD) *in vitro*.

Peak on p. 24 is misspelled as peck

Thanks for pointing out this mistake. We have corrected it in in the revised manuscript (please refer to Page 24 Line 35).

The methods section starting in the section "Cell culture and small interfering RNAs" through "Cell invasion" reads like a protocol rather than a description of what was actually done. This should be modified to reflect what was actually done.

Thanks for your suggestion. We have made modifications to this section in the revised manuscript. Please see Page 19 Line 15-Line 28 and Page 20 Line 40-Page 21 Line 22.

REFERENCE

1. Sun, Y. *et al.* Single-cell landscape of the ecosystem in early-relapse hepatocellular carcinoma. *Cell* **184**, 404-421 e16 (2021).
2. Hajdarovic, K.H. *et al.* Single-cell analysis of the aging female mouse hypothalamus. *Nat Aging* **2**, 662-678 (2022).
3. Garcia-Alonso, L. *et al.* Single-cell roadmap of human gonadal development. *Nature* **607**, 540-547 (2022).
4. Tirosh, I. *et al.* Single-cell RNA-seq supports a developmental hierarchy in human oligodendroglioma. *Nature* **539**, 309-313 (2016).
5. Treutlein, B. *et al.* Reconstructing lineage hierarchies of the distal lung epithelium using single-cell RNA-seq. *Nature* **509**, 371-5 (2014).
6. Miragaia, R.J. *et al.* Single-cell RNA-sequencing resolves self-antigen expression during mTEC development. *Sci Rep* **8**, 685 (2018).
7. Buenrostro, J.D. *et al.* Integrated Single-Cell Analysis Maps the Continuous Regulatory Landscape of Human Hematopoietic Differentiation. *Cell* **173**, 1535-1548 e16 (2018).
8. Yan, X., Orentas, R.J. & Johnson, B.D. Tumor-derived macrophage migration inhibitory factor (MIF) inhibits T lymphocyte activation. *Cytokine* **33**, 188-98 (2006).
9. Barbosa de Souza Rizzo, M. *et al.* Oral squamous carcinoma cells promote macrophage polarization in an MIF-dependent manner. *QJM* **111**, 769-778 (2018).
10. Simpson, K.D., Templeton, D.J. & Cross, J.V. Macrophage migration inhibitory factor promotes tumor growth and metastasis by inducing myeloid-derived suppressor cells in the tumor microenvironment. *J Immunol* **189**, 5533-40 (2012).
11. Yashar, W.M. *et al.* GoPeaks: histone modification peak calling for CUT&Tag. *Genome Biol* **23**, 144 (2022).
12. Kaya-Okur, H.S., Janssens, D.H., Henikoff, J.G., Ahmad, K. & Henikoff, S. Efficient low-cost chromatin profiling with CUT&Tag. *Nat Protoc* **15**, 3264-3283 (2020).
13. Kaya-Okur, H.S. *et al.* CUT&Tag for efficient epigenomic profiling of small samples and single cells. *Nat Commun* **10**, 1930 (2019).
14. Cheng, C., Min, R. & Gerstein, M. TIP: a probabilistic method for identifying transcription factor target genes from ChIP-seq binding profiles. *Bioinformatics* **27**, 3221-7 (2011).
15. Nguyen, D.D. *et al.* Integrative Bioinformatics and Functional Analyses of GEO, ENCODE, and TCGA Reveal FADD as a Direct Target of the Tumor Suppressor BRCA1. *Int J Mol Sci* **19**(2018).

Reviewers' Comments:

Reviewer #4:

Remarks to the Author:

The authors have adequately addressed the comments and this research is now suitable to be considered for publication.

Reviewer #5:

Remarks to the Author:

The authors have adequately addressed my prior comments.

I have additional minor comments regarding some framing of the results in the discussion.

"We found that in seminoma, a large proportion of immune cells are in a dysfunctional or exhausted state, while there is a significant part which have effective responses, indicating immune cells and tumor cells in a dynamic balance. However, based on the spatial distribution and proportion of tumors and immune cells, the seminoma TME seems to be in an immune-suppressed state, which could be a reason for certain seminoma cells to metastasize."

- "effective responses" should read "effector responses"

- there is no evidence that the dysfunctional/exhausted immune cells have anything to do with metastasis as these tumors were all localized - it would be fair to say that these cells could permit tumor cells to proliferate within the organ

"Additionally, we found that MIF is highly expressed in seminoma, and its receptor mainly expressed in the immune cells, indicating the tumor had frequent crosstalk to the immune cells in TME. Therefore, we propose MIF as a promising candidate target to treat seminoma, though additional studies are needed to further investigate its functional impact in seminoma and the underlying mechanism."

- The key question regarding MIF as a therapeutic target is whether metastatic seminoma expresses MIF. All localized samples including those in this cohort do not require any therapy beyond surgery.

REVIEWERS' COMMENTS

Reviewer #4 (Remarks to the Author):

The authors have adequately addressed the comments and this research is now suitable to be considered for publication.

We are appreciated to receive the positive feedback from the reviewer.

Reviewer #5 (Remarks to the Author):

The authors have adequately addressed my prior comments.

I have additional minor comments regarding some framing of the results in the discussion.

"We found that in seminoma, a large proportion of immune cells are in a dysfunctional or exhausted state, while there is a significant part which have effective responses, indicating immune cells and tumor cells in a dynamic balance. However, based on the spatial distribution and proportion of tumors and immune cells, the seminoma TME seems to be in an immune-suppressed state, which could be a reason for certain seminoma cells to metastasize."

- "effective responses" should read "effector responses"

Thanks for your suggestion. We have made modifications in the revised manuscripts. Please refer to Page 11 Line 33.

- there is no evidence that the dysfunctional/exhausted immune cells have anything to do with metastasis as these tumors were all localized - it would be fair to say that these cells could permit tumor cells to proliferate within the organ

Thanks for your comment. We modified the description in the revised manuscript. Please refer to Page 11 Line 36.

"Additionally, we found that MIF is highly expressed in seminoma, and its receptor mainly expressed in the immune cells, indicating the tumor had frequent crosstalk to the immune cells in TME. Therefore, we propose MIF as a promising candidate target to treat seminoma, though additional studies are needed to further investigate its functional impact in seminoma and the underlying mechanism."

- The key question regarding MIF as a therapeutic target is whether metastatic seminoma expresses MIF. All localized samples including those in this cohort do not require any therapy beyond surgery.

Thanks for your feedback. We agreed with the statement of the review regarding MIF being considered as a therapeutic target. We have found that MIF highly expressed in tumor cells within the tumor microenvironment and exerts inhibitory effects on the surrounding environment. This behavior itself is beneficial for tumor promotion. Therefore, we believed that if a tumor undergoes metastasis or intends to undergo metastasis, factors

like MIF were required to facilitate this process. To further validate, we examined the expression of *MIF* in a previously published data on seminoma that had metastasized to the lymph nodes ^[1]. Similarly, we observed that MIF highly expressed in tumor cells (Fig R5.1). Hence, we considered that *MIF*, even when found in situ within the tumor, can serve as a potential target for seminoma treatment.

Figure R5.1 MIF expression in seminoma metastasized to the lymph nodes.

1. Mo L, Yu Z, Lv Y, Cheng J, Yan H, Lu W, *et al.* Single-Cell RNA Sequencing of Metastatic Testicular Seminoma Reveals the Cellular and Molecular Characteristics of Metastatic Cell Lineage. **Front Oncol** 2022, 12: 871489.